# Cortico-thalamo-cortical interactions modulate electrically evoked EEG responses in mice

Leslie D Claar[1]*[†], Irene Rembado[1]*[†], Jacqulyn R Kuyat[1], Simone Russo[1,2], Lydia C Marks[1], Shawn R Olsen[1], Christof Koch[1]*

[1]MindScope Program, Allen Institute, Seattle, United States; [2]Department of Biomedical and Clinical Sciences "L. Sacco", University of Milan, Milan, Italy

**Abstract** Perturbational complexity analysis predicts the presence of consciousness in volunteers and patients by stimulating the brain with brief pulses, recording EEG responses, and computing their spatiotemporal complexity. We examined the underlying neural circuits in mice by directly stimulating cortex while recording with EEG and Neuropixels probes during wakefulness and isoflurane anesthesia. When mice are awake, stimulation of deep cortical layers reliably evokes locally a brief pulse of excitation, followed by a biphasic sequence of 120 ms profound *off* period and a *rebound* excitation. A similar pattern, partially attributed to burst spiking, is seen in thalamic nuclei and is associated with a pronounced late component in the evoked EEG. We infer that cortico-thalamo-cortical interactions drive the long-lasting evoked EEG signals elicited by deep cortical stimulation during the awake state. The cortical and thalamic off period and rebound excitation, and the late component in the EEG, are reduced during running and absent during anesthesia.

*For correspondence:
lesliec@alleninstitute.org (LDC);
irene.rembado@alleninstitute.org (IR);
christofk@alleninstitute.org (CK)

[†]These authors contributed equally to this work

## eLife assessment

This study makes a **fundamental** observation about the role of activity in the mouse thalamus on scalp recorded voltage fluctuations. The novel approach and sophisticated analysis of neural signals provides **compelling** support for the authors' observations. This work will likely be of broad interest to neuroscientists.

## Introduction

A long-standing clinical challenge has been to discover sensitive and specific biomarkers of consciousness. One obvious candidate is EEG, with high amplitude delta activity usually taken as an indicator of absence of consciousness, as during deep sleep, anesthesia, in vegetative state patients (also known as behavioral unresponsive wakefulness syndrome), or in coma (*Bai et al., 2017*; *Kobylarz and Schiff, 2005*; *Schiff et al., 2014*). However, given the vast diversity of patients and their etiology, spontaneous EEG can show very unusual spatiotemporal patterns, with attendant high false alarm and miss rates in diagnosing individual patients with disorders of consciousness (*Farisco et al., 2022*; *Frohlich et al., 2021*; *Thibaut et al., 2019*). More promising is the perturbational EEG (*Bai et al., 2021*), in which the brain is probed by a brief pulse (generated via transcranial magnetic stimulation [TMS] applied to the skull or electrical stimulation applied intracranially), and the resulting cortical activity is recorded using a high-density EEG electrode array or stereo-EEG electrodes (*Casali et al., 2013*; *Casarotto et al., 2016*; *Comolatti et al., 2019*; *Massimini et al., 2005*; *Pigorini et al., 2015*; *Rosanova et al., 2018*). A simple algorithm then computes the *perturbational complexity index* (PCI) of the brain's reverberations to this pulse from which the presence of consciousness can be inferred with

unprecedented sensitivity (low false alarm rate) and specificity (low miss rate), at the level of individual patients (*Bai et al., 2021*; *Kondziella et al., 2020*).

PCI has been comprehensively studied and validated in humans since its introduction by Massimini and colleagues in 2005 (*Casali et al., 2013*; *Casarotto et al., 2016*; *Comolatti et al., 2019*; *Ferrarelli et al., 2010*; *Massimini et al., 2005*; *Rosanova et al., 2018*). *Pigorini et al., 2015* proposed that during non-rapid eye movement (NREM) sleep, when the brain is characterized by cortical bistability, neurons tend to fall into a down-state following activation (either endogenous or exogenous) preventing extended causal interactions that are typical during wakefulness (*Hill and Tononi, 2005*; *Rosanova et al., 2018*; *Timofeev et al., 2001*).

Limitations in access to high resolution cellular recordings in humans have motivated recent efforts to translate the PCI technique to model systems for more detailed studies of the circuit mechanisms underlying complexity changes. In 2018, D'Andola et al. demonstrated that an approximation of PCI developed for in vitro measurements captured complexity differences in ferret cortical slices exhibiting sleep- and awake-like activity patterns. They showed that during a sleep-like state, electrical stimulation induced a down-state that disrupted the complex pattern of activation observed in the awake-like state (*D'Andola et al., 2018*). *Arena et al., 2021* were the first to recapitulate the human PCI results in vivo in rodents; they computed the complexity of electrically evoked EEG responses in rats and showed that PCI was high during wakefulness and low during anesthesia. In their comprehensive study, they found that in the anesthetized state (via propofol or sevoflurane), stimulation was followed by a widespread suppression of high frequencies in the EEG responses, suggestive of a down-state, in agreement with previous findings (*D'Andola et al., 2018*; *Pigorini et al., 2015*). *Dasilva et al., 2021* measured PCI in vivo in anesthetized mice and showed that complexity can be modulated even within the anesthetized state. They showed that PCI was highest for mice under light isoflurane anesthesia (defined as a concentration of 0.1%), decreasing systematically at medium (0.34%) and deep (1.16%) concentrations. Recently, *Cavelli et al., 2022* validated the use of PCI in freely moving rodents across wakefulness, both REM and NREM sleep, and anesthesia by demonstrating that PCI calculated on cortical laminar local field potential (LFP) signals decreases in unconscious states (both natural and induced) in rats and mice. While pioneering, none of these studies simultaneously recorded micro- and macro-signals – neurons and LFP at hundreds of sites, together with EEG.

To investigate how cortico-cortical and cortico-thalamic activity influences the complexity of the evoked responses, we used EEG simultaneously with Neuropixels probes (*Jun et al., 2017*) to record brain-wide evoked responses to cortical electrical stimulation in head-fixed mice that were awake and, subsequently, anesthetized with isoflurane. Due to their ability to record spiking and LFP signals from hundreds of sites across cortical layers and subcortical structures at 20 μm resolution, Neuropixels probes provide unprecedented access to the intra- and interareal dynamics that underlie the macro-scale EEG signals. We show that cortical stimulation elicits a widespread, complex event-related potential (ERP) in the EEG signals in the awake state, but a much simpler ERP in the isoflurane-anesthetized state, in agreement with what has been shown in humans (*Casali et al., 2013*) and rodents (*Arena et al., 2021*; *Cavelli et al., 2022*). We demonstrate a stereotyped pattern of activity to stimulation in deep (but not superficial) cortical layers – brief excitation, followed by a profound *off* period and a *rebound* excitation ('rebound' is a term often used to describe a period of enhanced spiking following a period of inhibition or silence; *Grenier et al., 1998*; *Guido and Weyand, 1995*; *Roux et al., 2014*). This sequence is repeated in the thalamus, supported by burst firing in excitatory thalamic neurons. Based on relative timing between cortical and thalamic evoked activity, we infer that thalamic bursting is necessary for the late, evoked EEG component seen in response to electrical stimulation, a novel result that links the ERP to activity in the cortico-thalamo-cortical (CTC) loop.

## Results

### Global evoked responses are modulated by the depth of the cortical stimulation site

We recorded global EEG-like neural signals using a multi-electrode surface array on the skull, but below the scalp (*Jonak et al., 2018*; *Land et al., 2019*), in head-fixed mice. The multi-electrode array consisted of 30 electrodes situated over primary and secondary motor, somatosensory, visual, and retrosplenial areas in both hemispheres (*Figure 1A*). We used an average reference montage that

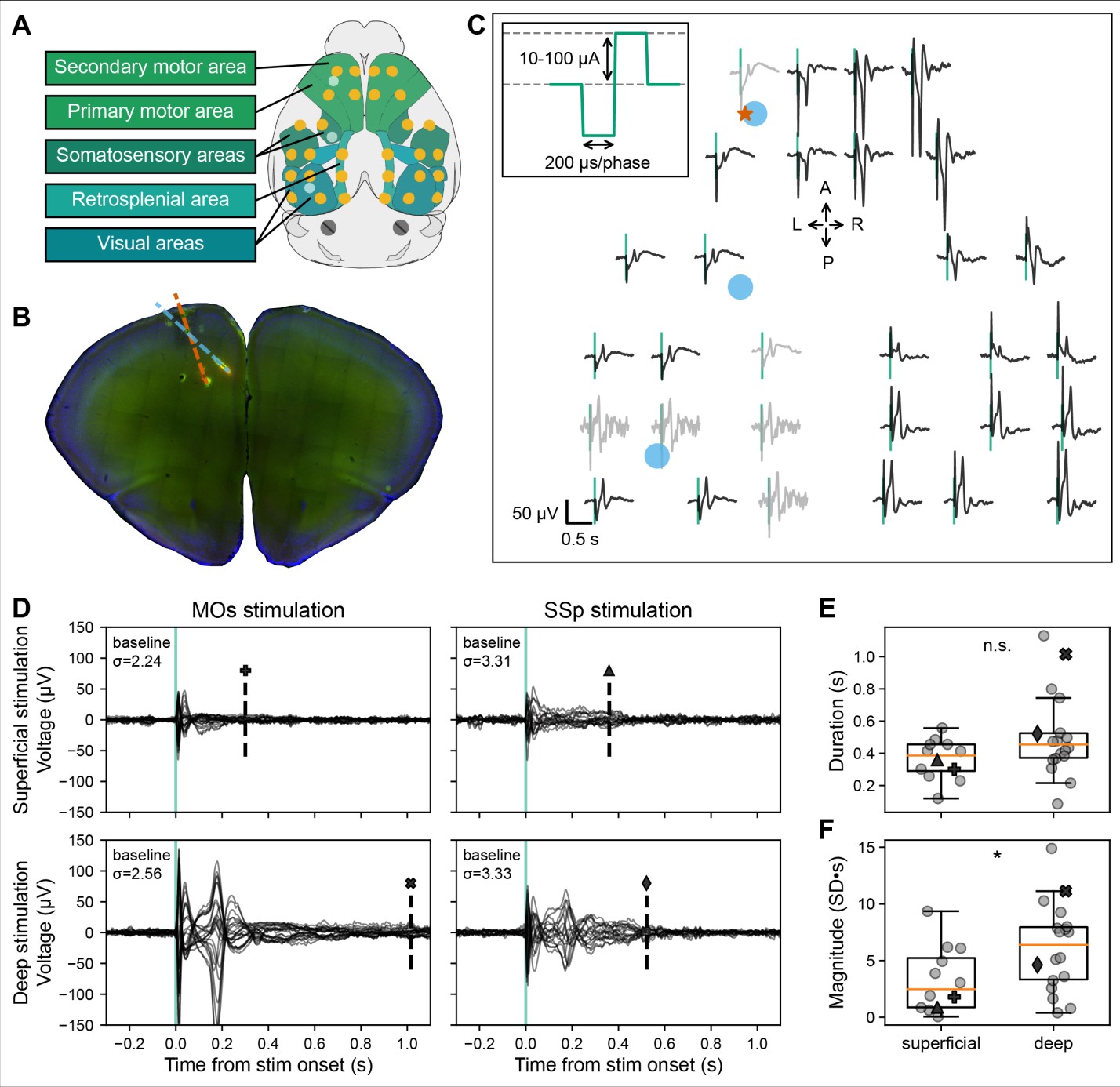

**Figure 1.** Evoked EEG responses to single pulse electrical stimulation in awake, head-fixed mice. (**A**) Schematic of the 30-channel surface array (yellow circles) implanted on top of the skull over major brain areas: motor, somatosensory, retrosplenial, and visual areas (schematic created using *brainrender*; *Claudi et al., 2021*). The circular, platinum EEG electrodes are 500 μm in diameter. The three light blue circles correspond to the locations of the three acute craniotomies to place up to three Neuropixels probes and the bipolar stimulating electrode. The schematic also shows two skull screws over the cerebellum that serve as the reference and the ground for the EEG signals. (**B**) Histological image of a coronal brain slice showing the location of the bipolar stimulating electrode (with tips in secondary motor area [MOs], layer 5; red dashed line) and one of the Neuropixels probes (spanning layers of motor and anterior cingulate areas; blue dashed line) with fluorescent dyes (that appear red and green in the image). (**C**) Evoked responses from each of the 30 EEG electrodes from the awake, head-fixed mouse from –0.2 to +0.8 s following the electrical stimulus (vertical green line marks the onset time). Traces are arranged in the approximate orientation of the EEG array over the skull surface. Traces in black and gray represent signals that did and did not pass a quality control step, respectively. The red star and blue circles mark the approximate insertion point of the bipolar stimulating electrode and the Neuropixels probes, respectively. **Inset:** Single current pulses were biphasic (200 μs/phase), charge-balanced, and cathodic-first,

*Figure 1 continued on next page*

*Figure 1 continued*

with a current amplitude between 10 and 100 µA. (**D**) Event-related potential (ERP; –0.3 to +1.1 s around stimulus onset) with all EEG electrode traces superimposed (butterfly plots). Each of the four panels represents data from a different stimulated area and depth: top and bottom left – superficial and deep layer (same as in panel C) MOs stimulation in the same subject; top and bottom right – superficial and deep layer primary somatosensory area (SSp) stimulation in a different subject. The dashed vertical line indicates the duration of the evoked signal; the marker above matches with the marker representing the value in panels **E** and **F**. The 'baseline σ' indicates the SD (in µV) over all electrodes during the 2 s preceding the stimulus. (**E**) Duration of the ERPs for all subjects based on the stimulation depth: superficial (N=12) vs. deep (N=18). (**F**) Normalized magnitude of the ERPs for all subjects based on the stimulation depth: superficial (N=12) vs. deep (N=18). For further details, see method 'ERP duration and magnitude' and *Figure 1—figure supplement 1*. Boxplots show median (orange line), 25th, and 75th percentiles; whiskers extend from the box by 1.5× the inter-quartile range (IQR). Student's two-tailed t-test; * weak evidence to reject null hypothesis (0.05>p>0.01), ** strong evidence to reject null hypothesis (0.01>p>0.001), and *** very strong evidence to reject null hypothesis (0.001>p).

The online version of this article includes the following figure supplement(s) for figure 1:

**Figure supplement 1.** Demonstration of duration and magnitude calculation for evoked responses.

**Figure supplement 2.** Spatial extent of evoked EEG responses in awake, head-fixed mice.

removed signals common to all EEG electrodes (see Methods). We inserted a bipolar wire electrode intra-cortically to repeatedly deliver a single electrical current pulse into the cortex and measured the evoked potentials with the EEG array (*Figure 1B and C*). The current pulses were biphasic (200 µs/phase) with an amplitude between 10 and 100 µA (details in *Figure 1C* inset). Stimulation artifacts in the EEG signals were reduced by replacing the signal between 0 and +2 ms following each stimulus with the signal between –2 and 0 ms; this was done to all traces in all trials during an offline, signal pre-processing step. All subsequent analyses of the ERP (the trial-averaged EEG response) exclude the signal between –2 and +2 ms. During the experiment, we closely observed the animal for signs of electrically evoked motor twitches and chose lower stimulation amplitudes if we observed any.

To understand how the features of the ERP depend on the location of the electrical stimulation, we varied both the area and the depth (or cortical layer) of the stimulating electrode. We stimulated in the secondary motor area (MOs) or in the primary somatosensory area (SSp) in layer 2/3 (superficial: 0.41±0.04 mm below the brain surface) or in layer 5/6 (deep: 1.06±0.05 mm below the brain surface; *Figure 1D*). We found that regardless of area, when we stimulated layer 5/6 during the awake period, we usually observed two prominent peaks in the ERP: an initial response around 25 ms and a secondary peak at around 180 ms post-stimulation (*Figure 1C and D*).

The early component was preserved, whereas the second, late component was not evident when stimulating superficially in either area (*Figure 1D*). The total duration of the ERP was shorter when stimulating superficial layers compared to deep layers although not significant (mean ERP duration for superficial stimulation: 0.4±0.0 s; deep stimulation: 0.5±0.1 s; Student's two-tailed t-test, p=0.0843; *Figure 1E*, *Figure 1—figure supplement 1*). When stimulating superficially, ERPs had a significantly smaller normalized magnitude compared to stimulating deep layers (mean ERP magnitude for superficial stimulation: 3.3±0.8 SD·s; deep stimulation: 6.2±0.9 SD·s; Student's two-tailed t-test, p=0.0354; *Figure 1F*, *Figure 1—figure supplement 1*).

The amplitude of the first peak in the ERP decays systematically when moving away from the stimulation site until it eventually flips its sign, most likely reflecting volume conduction (*Figure 1—figure supplement 2*). The magnitude and polarity of the second component likewise changes continuously but in a different pattern, suggesting a different mechanistic origin.

## The spiking response pattern of the stimulated cortex echoes the ERP

In addition to recording EEG signals, we simultaneously collected data from up to three Neuropixels probes – linear silicon probes with a 10 mm long non-tapered shank with 384 simultaneously recorded electrodes capable of capturing LFP and action potentials (*Jun et al., 2017*). The Neuropixels probes were placed in such a manner as to record from cortex (motor MO, anterior cingulate ACA, somatosensory SS, and visual VIS) and sensorimotor-related thalamic nuclei (SM-TH, *Figure 2A*), see full list of thalamic nuclei in the Methods section (*Guo et al., 2017*; *Harris et al., 2019*; *Hooks et al., 2013*). The Neuropixels probes were inserted approximately perpendicular to the cortical surface. This allowed us to observe the LFP and the spiking activity of hundreds of individual cortical and thalamic neurons. From the LFP, we inferred the current source density (CSD) using a computational method that assumes an ohmic conductive medium, constant extracellular conductivity (0.3 S/m),

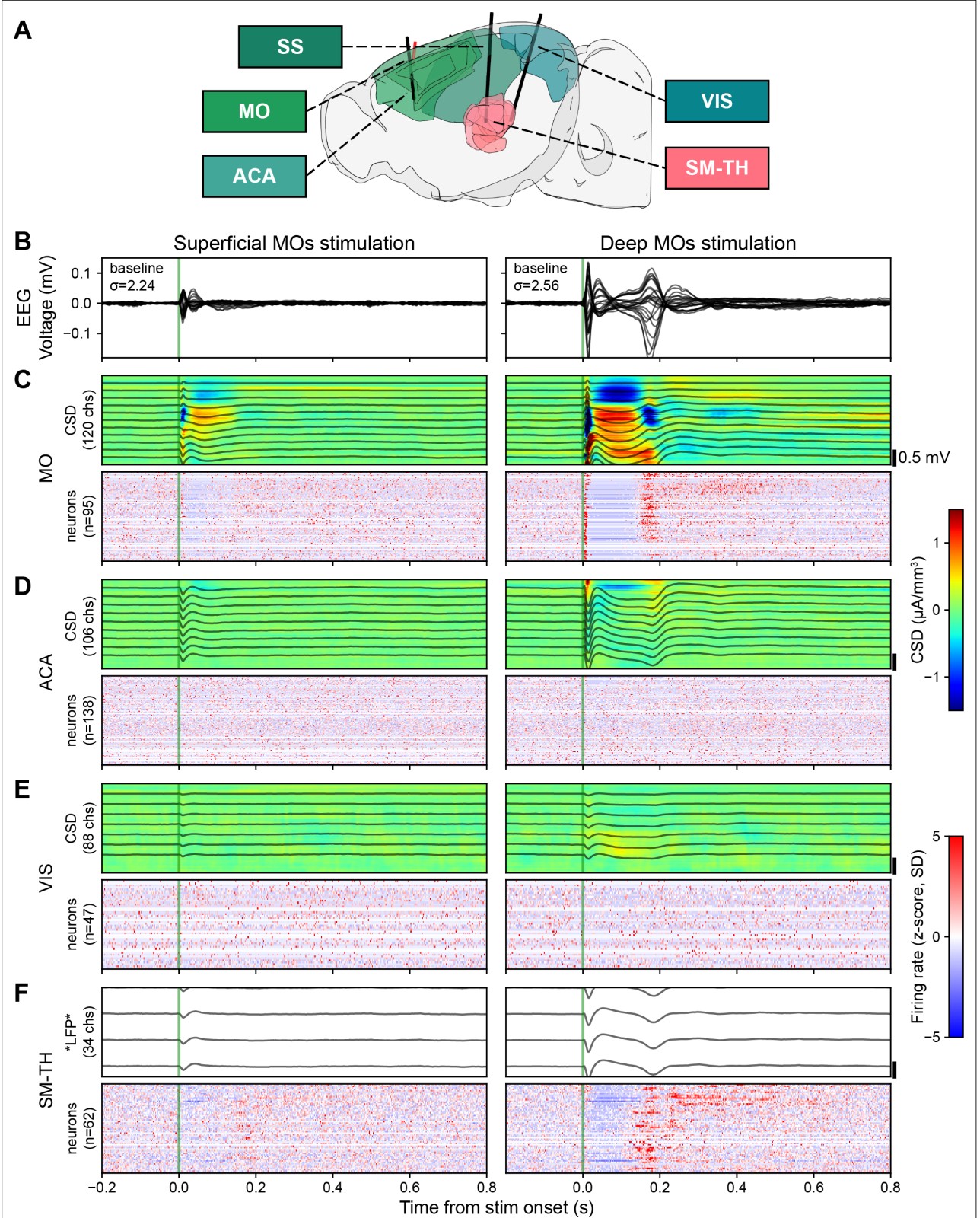

**Figure 2.** Electrical stimulation evokes strong responses in the EEG, LFP, CSD, and in some populations of neurons (locally in MO and in SM-TH) when deep layers of MOs are directly activated. (**A**) Sagittal schema of the mouse brain, highlighting motor (MO), anterior cingulate (ACA), somatosensory (SS), visual (VIS), and somatomotor-related thalamic (SM-TH) areas (created using *brainrender*; **Claudi et al., 2021**). Solid black lines show the approximate locations of three acutely inserted Neuropixels probes; the red line indicates the stimulating electrode in the deep layers of MOs. (**B**)

*Figure 2 continued on next page*

*Figure 2 continued*

Butterfly plots of the event-related potential (ERP; –0.2 to +0.8 s around stimulus onset) evoked during the awake state. Each column represents data from a different stimulated depth (superficial and deep MOs) in the same subject (same subject shown in *Figure 1D* left top and bottom). (C) Evoked responses from the Neuropixels electrodes in MO. (**Top**) From the measured LFP band (black traces representing 1 out of every 10 channels), the CSD response was computationally inferred (heat map, red and blue represent sources and sinks, respectively). The number of Neuropixels channels used to compute CSD is indicated along the y-axis. Bottom: Normalized firing rate, reported as a z-score of the average, pre-stimulus firing rate, of all neurons (only regular spiking [RS] neurons in cortical regions) recorded by the Neuropixels probes targeting the area of interest. The number of neurons (**n**) in each area is included along the y-axis. (**D**) Evoked responses from ACA, same as panel C. (**E**) Evoked responses from VIS, same as panel C. (**F**) Evoked responses from SM-TH, CSD was not computed for SM-TH because thalamic structures, unlike cortex, do not contain oriented neural elements; therefore, it would not be interpretable.

and homogeneous in-plane neuronal activity, with the boundary condition of zero current outside the sampled area (*Freeman and Nicholson, 1975*; *Mitzdorf, 1985*).

We inserted a Neuropixels probe near the stimulating electrode (within 0.5 mm) and, often, up to two additional Neuropixels probes at distal locations. We observed direct responses (i.e. defined operationally as neurons that spike between 2 and 25 ms following the electrical pulse; this might miss a handful of very rapidly (i.e.<2 ms) responding neurons; see *Sombeck et al., 2022*; *Stoelzel et al., 2009*) as well as indirect responses to the electrical stimulation. To look for cell-type specific activation patterns, we classified regular spiking (RS) and fast spiking (FS) neurons (putative pyramidal and putative parvalbumin-positive inhibitory neurons, respectively) based on their spike waveform duration (RS duration >0.4 ms; FS duration ≤0.4 ms; *Barthó et al., 2004*; *Bortone et al., 2014*; *Bruno and Simons, 2002*; *Jia et al., 2019*; *Niell and Stryker, 2008*; *Sirota et al., 2008*).

Stimulating superficial MOs evoked a local spike, LFP and CSD response, followed by a period of quiescence lasting 94.2±16.1 ms in MO neurons (*Figure 2C*, left). There were minimal to no evoked responses (LFP, CSD, or spikes) in other sampled areas (*Figure 2D–F*, left).

Stimulating deep MOs evoked a robust change in LFP and CSD, accompanied by spiking activity reflective of the two components in the ERP (*Figure 2B* right): an initial excitation within 25 ms (peak population firing rate 38.1±4.2 Hz), followed by quiescence (duration 125.0±5.5 ms) and a longer period of strong excitation (peak population firing rate 7.9±0.7 Hz; *Figure 2C*, right). This cortical response pattern was quite stereotyped – an initial excitation, followed by an off period and a strong rebound excitation – and was also observed in SM-TH (*Figure 2F*, right). We seldom observed this pattern for superficial stimulation (*Figure 2C*, left). Indeed, on average, about three times more cortical neurons respond significantly to deep than to superficial stimulation (Figure 4A and D).

## Global ERPs are associated with widespread evoked cortical activity

Intrigued by the apparent lack of firing in non-stimulated cortical regions, we quantified the evoked magnitude of the LFP, CSD, and spiking in response to superficial and deep stimulation (note that our magnitude metric accounts for both reductions as well as increases and is calculated as the integral of the z-score relative to background; see method "LFP, CSD, and population spiking magnitude"; *Figure 3A–C*). The magnitude of the LFP near the stimulation site is greater than background activity (54.0 SD·s in MO) and decays with distance from the stimulation site (18.0 SD·s in VIS; *Figure 3A*). Because the LFP reflects the summation of local presynaptic activity as well as volume conduction from more distant current sources (*Kajikawa and Schroeder, 2011*), we examined the CSD responses, which minimize the effect of volume conduction and emphasize local current flows. The magnitude of the CSD shows a similar pattern: large in the stimulated cortex (51.4 SD·s in MO) and smaller at more distant sites (4.8 SD·s in VIS), though the response is larger than baseline in both (*Figure 3B*). Yet while we see robust evoked firing of cortical neurons (30.9 SD·s in MO) close to the stimulation electrode, changes in firing are often much smaller in distant regions (1.8 SD·s in VIS; *Figure 3C*).

Stimulating superficial layers significantly modulated local LFP, CSD, and cortical spiking with respect to background (N=6; LFP median 19.8 [16.3–24.2 inter-quartile range, IQR], Wilcoxon signed-rank test corrected for multiple comparisons using Benjamini-Hochberg false discovery rate, p=0.0313; CSD 20.8 [17.5–22.9 IQR], p=0.0313; spiking 1.8 [0.5–5.0 IQR], p=0.0313; *Figure 3D*, left bars). It also modulated LFP and CSD in non-stimulated cortical areas (N=8; LFP 7.5 [5.1–9.7 IQR], p=0.0078; CSD 3.9 [2.7–10.0 IQR], p=0.0078), though to a lesser extent compared to the stimulated

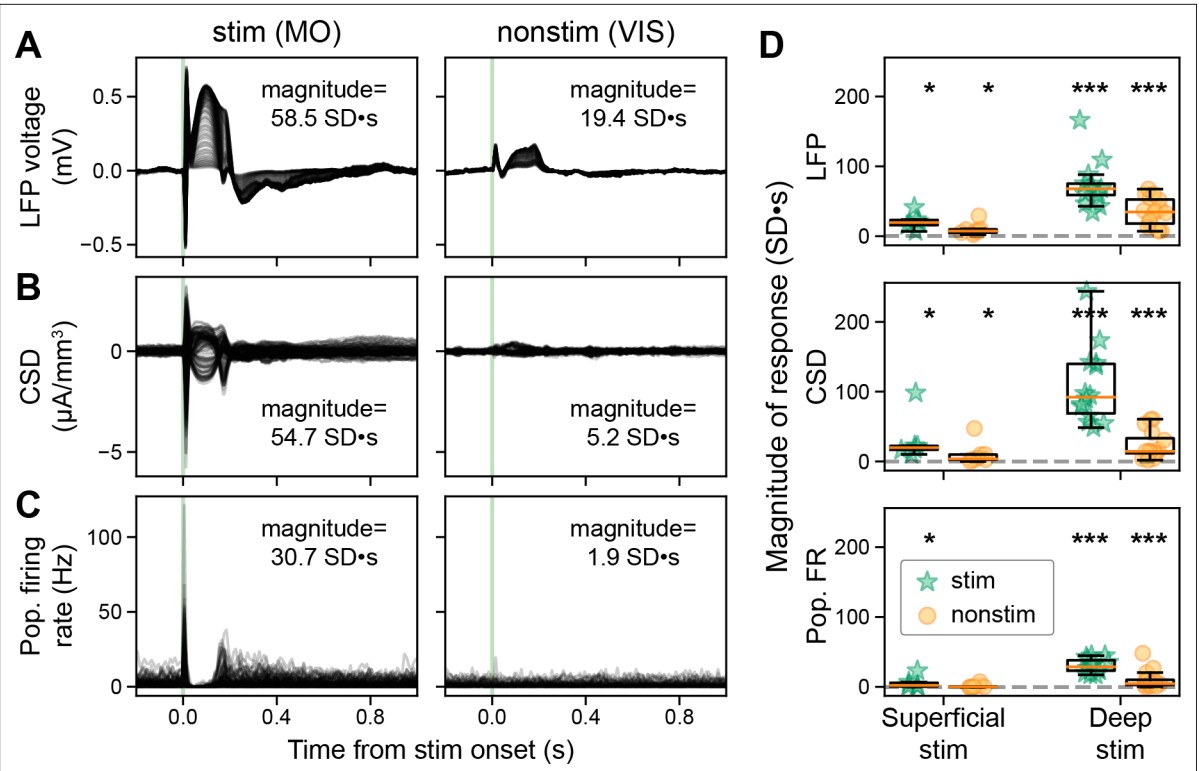

**Figure 3.** Direct stimulation of deep, but not superficial, cortical layers evokes widespread responses. (**A**) Evoked local field potential (LFP; –0.2 to +1.2 s around stimulus onset) from deep MOs stimulation (same subject as *Figure 1D* bottom left), all recorded cortical sites along the Neuropixels shaft are superimposed. The two columns represent data from stimulated (left) and a non-stimulated cortex (right). The magnitude is the integrated area of the response (0 to +0.5 s from stimulus onset), z-scored relative to the integrated area of the background activity calculated by shuffling the stimulation onset 1000 times. (**B**) Evoked current source density (CSD) from deep MOs stimulation, all recorded cortical sites along the Neuropixels shaft are superimposed. (**C**) Evoked neuronal firing rates from deep MOs stimulation, each trace represents a single RS neuron. (**D**) Magnitude of the evoked LFP (top), CSD (middle), and population spiking (bottom) to superficial (left) and deep stimulation (right) for stimulated and non-stimulated cortical regions (green stars and orange circles, respectively), dashed gray line at zero. Superficial stimulation: n=6 stimulated regions in N=6 mice and n=17 non-stimulated cortical regions in N=7 mice. Deep stimulation: n=15 stimulated regions in N=15 mice and n=35 non-stimulated cortical regions in N=16 mice. Boxplots show median (orange line), 25th, and 75th percentiles; whiskers extend from the box by 1.5× the IQR. Wilcoxon signed-rank test, corrected for multiple comparisons using Banjamini-Hochberg false discovery rate; * weak evidence to reject null hypothesis (0.05>p>0.01), ** strong evidence to reject null hypothesis (0.01>p>0.001), and *** very strong evidence to reject null hypothesis (0.001>p).

area. We did not detect evoked spiking outside of the stimulated area for superficial stimulation (–0.2 [−0.5–0.3 IQR], p=0.7422).

When we stimulated deep layers, we again observed large evoked LFP, CSD, and cortical spiking (deep, N=14; LFP 68.5 [55.3–79.2 IQR], Wilcoxon signed-rank test corrected for multiple comparisons using Benjamini-Hochberg false discovery rate, p=1.221E-4; CSD 95.1 [66.9–139.8 IQR], p=1.221E-4; spiking 29.3 [22.4–38.8 IQR], p=1.221E-4; *Figure 3D*, right bars). In contrast to superficial stimulation, deep stimulation-evoked responses in non-stimulated cortical regions were significantly greater than background (LFP 34.2 [18.5–51.3 IQR], p=3.052E-5; CSD 14.9 [11.9–33.4 IQR], p=3.052E-5, spiking 4.4 [2.7–10.1 IQR], p=3.052E-5).

Together, these data show that cortical stimulation evoked widespread changes in the LFP and CSD, but only evoked ipsilateral cortical spiking outside of the stimulated area when stimulating deep layers.

To further investigate the evoked neural spiking across both stimulated and non-stimulated areas, we computed the percentage of modulated (significantly increased or decreased spiking relative to the pre-stimulus baseline) RS neurons per subject in three temporal windows following the stimulus: 2–25 ms (initial excitation), 25–150 ms (off period), and 150–300 ms (rebound excitation; *Figure 4*). The three temporal windows were chosen to match the stereotyped firing patterns, which align with maximum peaks in the ERPs and the period in between (*Figure 1—figure supplement 2*). We

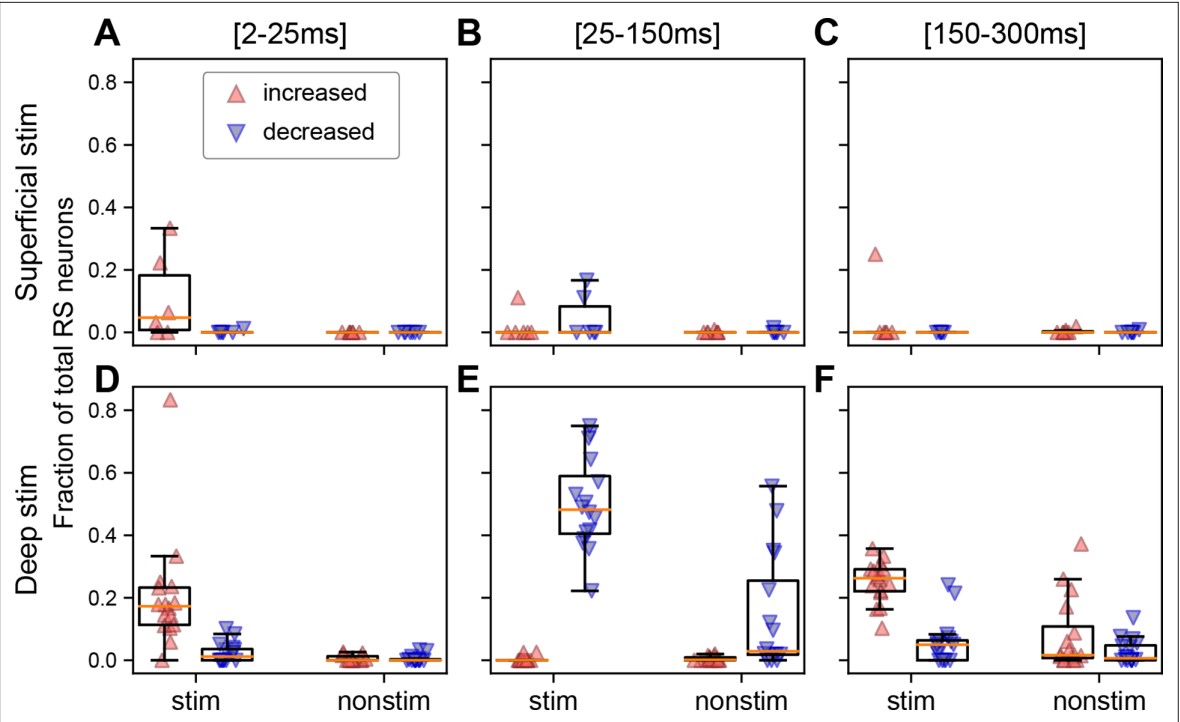

**Figure 4.** The fraction of the neuron population per ipsilateral cortical area in each subject that is significantly modulated depends on the depth of stimulation and the temporal window. (**A**) Fraction of regular spiking (RS) neurons per cortical area in each subject that exhibit a significantly increased (red upward triangle) or decreased (blue downward triangle) response in the first 25 ms for superficial stimulation. Fraction of RS neurons that exhibit a significant increase or decrease (**B**) 25–150 ms or (**C**) 150–300 ms following the stimulus. (**D–F**) Same as panels **A–C** but for deep stimulation. Panels **A–C** are derived from 870 RS neurons from stimulated cortex and 4110 RS neurons from non-stimulated cortical regions in N=7 mice and panels **D–F** from 2559 RS neurons from stimulated cortex and 9363 RS neurons from non-stimulated cortical regions in N=17 mice. Boxplots show median (orange line), 25th, and 75th percentiles; whiskers extend from the box by 1.5× the IQR.

The online version of this article includes the following figure supplement(s) for figure 4:

**Figure supplement 1.** The fraction of the FS neuron population that is significantly modulated depends on the depth of stimulation and the temporal window.

separately analyzed FS neurons and found they behave like RS neurons (***Figure 4—figure supplement 1***); therefore, further analyses focus only on RS neurons.

During the initial excitation, the fraction of activated (significantly increased spiking relative to baseline) neurons was always higher in the stimulated cortex than other cortical areas, for both superficial and deep stimulation. The fraction of activated neurons in non-stimulated ipsilateral cortical regions was only marginally above zero (***Figure 4A and D***). Indeed, across all 24 animals, only an average of 0.5% (minimum 0% and maximum 2.6%) of the neurons was modulated in the initial excitation, compared to an average of 17.8% (minimum 0% and maximum 83.3%) in stimulated cortex.

Next, we quantified the fraction of modulated RS neurons in the off period between 25 and 150 ms, capturing the period between the two large ERP amplitude components (present with deep stimulation; ***Figure 1—figure supplement 2***). There were very few neurons with increased spiking, regardless of stimulation depth and area (mean 0.4%, minimum 0%, and maximum 2.8%; ***Figure 4B and E***). In contrast, spiking in 18/22 subjects was significantly reduced from baseline following either superficial or deep stimulation. Interestingly, some non-stimulated areas showed decreased spiking but only in response to deep stimulation (mean 10.1%, minimum 0%, and maximum 55.7%).

The final temporal window we considered was the rebound excitation window between 150 and 300 ms, coinciding with the second component in the ERP. All 16 subjects who received deep stimulation showed significant rebound excitation in the stimulated cortex, whereas only 1/6 superficial stimulation subjects did (***Figure 4C and F***). Some subjects who received deep stimulation showed significant rebound excitation in the non-stimulated areas (6/16; ***Figure 4F***), but it was nonexistent amongst the superficial stimulation subjects (0/7; ***Figure 4C***).

To summarize, deep cortical stimulation is more likely to significantly modulate single neurons outside of the stimulated cortical area. This largely consisted of decreases in firing in the 25–150 ms window and mixed increases and decreases in firing during the 150–300 ms window (**Figure 4F**).

## CTC interactions underlie features of the ERPs

The striking triphasic cortical spiking pattern for deep stimulation is replicated in the associated thalamic nuclei that are connected to the stimulated cortical area (here, SM-TH; **Guo et al., 2017**; **Harris et al., 2019**; **Hooks et al., 2013**; bottom right in **Figure 2F**), though shifted in time. Specifically, we observed a brief excitation (peak population firing rate 22.3±3.2 Hz), a 74.6±15.5 ms long period of suppression, followed by rebound excitation (peak population firing rate 14.7±1.0 Hz). This is not the case when stimulating superficial layers (**Figure 2F**, bottom left).

We quantified the fraction of significantly modulated neurons in the SM-TH population and found that the neurons were almost eight times more likely to be modulated by deep than superficial

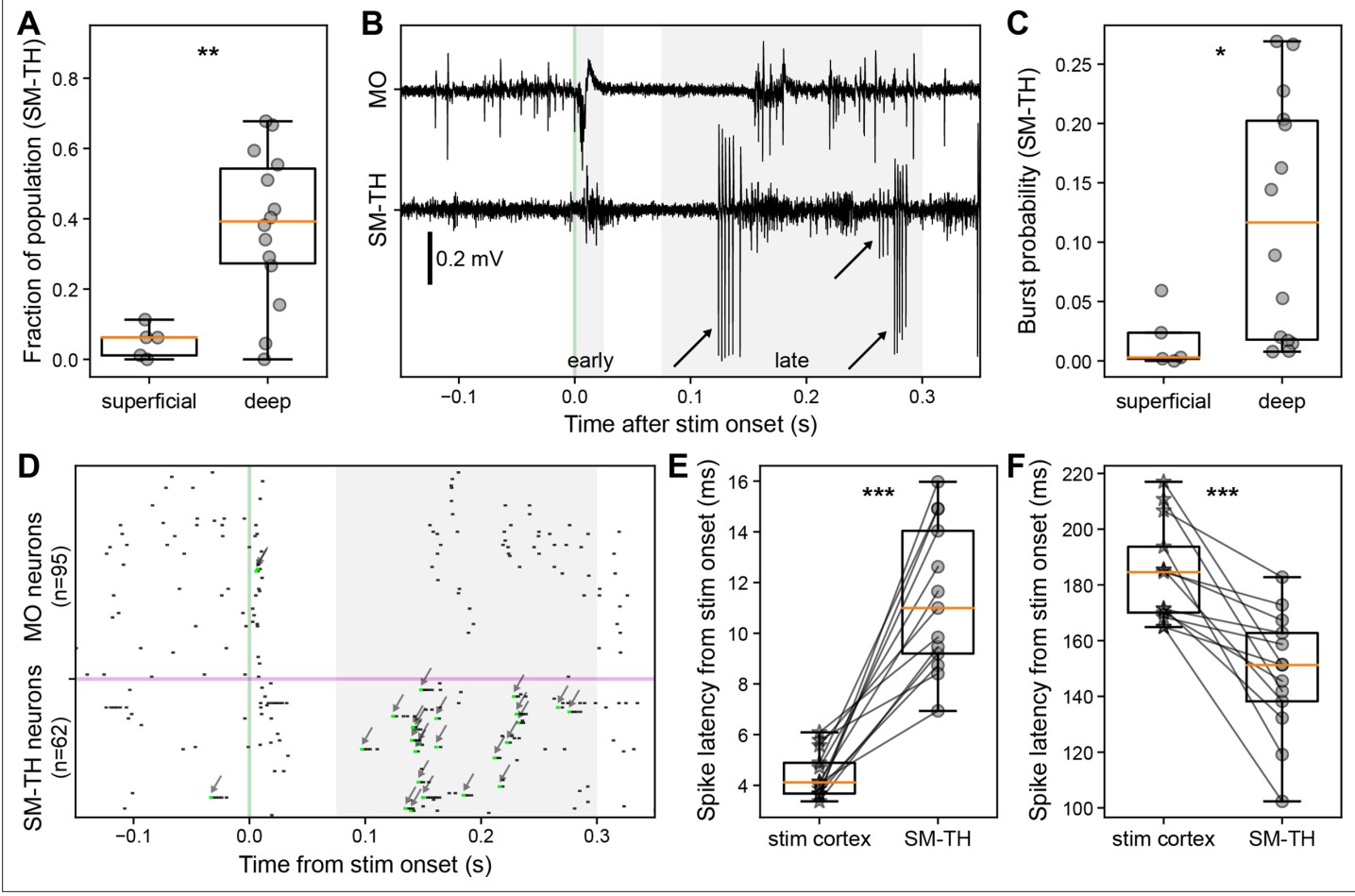

**Figure 5.** Cortical and thalamic neural dynamics evoked by cortical electrical stimulation. (**A**) Fraction of somatomotor-related thalamic (SM-TH) neurons that exhibit a significant increase or decrease in firing rate compared to baseline between 2 ms and 300 ms following stimulus onset for superficial (N=5) and deep stimulation (N=14). (**B**) Raw traces from the Neuropixels spike band data for motor (MO; top) and SM-TH (bottom; –0.15 to +0.35 s) for one exemplar trial with deep MOs stimulation. Action potential bursts in the SM-TH are flagged with arrows. (**C**) Probability (fraction of total trials) that the SM-TH produces bursts within 75–300 ms from stimulus onset for superficial (N=5) and deep stimulation (N=14). (**D**) Single trial raster plot showing spiking of MO (top) and SM-TH neurons (bottom) in response to a single deep electrical pulse (at the green vertical line). Action potential bursts are flagged with arrows. (**E**) Latency to first-spike (2–25 ms) for responsive RS neurons in stimulated cortex (stars) and in associated SM-TH (circles). Populations are recorded simultaneously in each subject, represented by the connecting black lines (N=13). (**F**) Latency to spike in the late window (75–300 ms) for responsive RS neurons in stimulated cortex (stars) and in SM-TH (circles), as in panel **E**. Boxplots show median (orange line), 25th, and 75th percentiles; whiskers extend from the box by 1.5× the IQR. Student's two-tailed t-test or paired t-test (for normally distributed data), or Mann-Whitney U test (non-parametric); * weak evidence to reject null hypothesis (0.05>p>0.01), ** strong evidence to reject null hypothesis (0.01>p>0.001), and *** very strong evidence to reject null hypothesis (0.001>p).

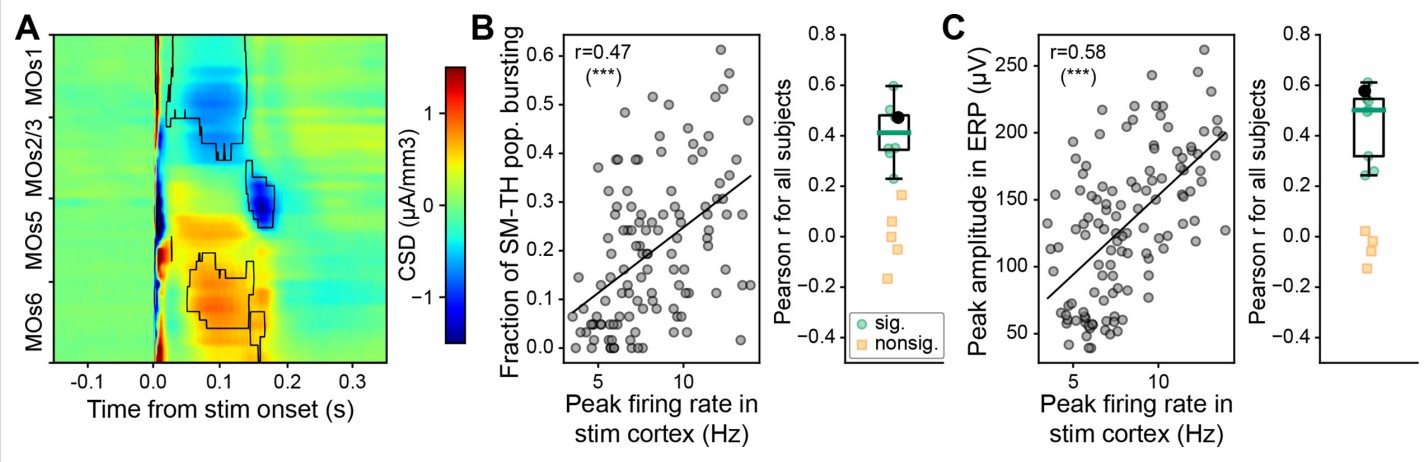

**Figure 6.** Thalamic origins of the late ERP component. (**A**) Population level (N=9) current source density (CSD) analysis of MOs after deep MOs stimulation (sinks blue and sources red). The black traces outline the areas with significant and consistent responses across subjects (Wilcoxon signed-rank test, p<0.05). (**B**) Left: Pearson correlation between fraction of somatomotor-related thalamic (SM-TH) neurons that burst and peak cortical population firing rate on a trial-by-trial basis for one mouse (same subject as in *Figure 1D* bottom left). Right: Same as left, computed for all (N=13) mice with deep stimulation; the black circle represents the subject from the left panel, green circles represent subjects with a significant correlation (p<0.05), and yellow squares represent subjects with a non-significant correlation. (**C**) Left: Pearson correlation between peak amplitude of the second, late component in the event-related potential (ERP) and peak cortical population firing rate on a trial-by-trial basis for one example mouse (as in panel **B**). Right: Same as left, computed for all subjects with deep stimulation (N=13), represented as in panel **B**. Boxplots show median (green line), 25th, and 75th percentiles; whiskers extend from the box by 1.5× the IQR. * Weak evidence to reject null hypothesis (0.05>p>0.01), ** strong evidence to reject null hypothesis (0.01>p>0.001), and *** very strong evidence to reject null hypothesis (0.001>p).

The online version of this article includes the following figure supplement(s) for figure 6:

**Figure supplement 1.** ERP is more correlated to CSD of deep, rather than superficial, cortical layers.

**Figure supplement 2.** The amplitude of the second, late component in the ERP is correlated with SM-TH bursting, but not the population firing rate.

stimulation (superficial: 5.0±4.1% of SM-TH population, N=5; deep: 37.9±20.6%, N=14; Student's two-tailed t-test, p=0.0037; *Figure 5A*).

Thalamic response shows bursts of action potentials during the rebound (*Figure 5B and D*); bursting is a well-known activity mode of thalamic relay cells and is defined as two or more consecutive spikes with an inter-spike interval less than 4 ms preceded by at least 100 ms of silence (*Contreras and Steriade, 1995*; *Grenier et al., 1998*; *Guido and Weyand, 1995*; *Halassa et al., 2011*; *Lu et al., 1992*; *Nestvogel and McCormick, 2022*). We quantified the average burst probability (75–300 ms after stimulus onset) and found that deep stimulation had a significantly higher likelihood of evoking bursts (superficial stimulation: median 0 [0–0.02 IQR]; deep stimulation: median 0.12 [0.02–0.2 IQR]; Mann-Whitney *U* test, p=0.0258; *Figure 5C*).

We hypothesized that deep stimulation elicits spiking in cortico-thalamic projection neurons, followed – with some delay due to axonal propagation and synaptic transmission – by thalamic neurons, whereas during the rebound period thalamic bursting precedes cortical rebound spiking (*Figure 5B and D*). Indeed, during the initial excitation, stimulated cortical RS cells had a median first-spike latency of 4.4±0.9 ms, with the thalamic population following at 11.4±2.8 ms (mean latency difference of 6.9 ms; paired t-test, p=4.838E-6; *Figure 5E*). During the rebound excitation window, the temporal relationship flipped: the thalamic population had a median rebound spike latency of 148.2±21.2 ms, with cortex following at 184.2±17.3 ms (mean latency difference −36.0 ms; paired t-test, p=2.136E-4; *Figure 5F*).

Similarly, the CSD analysis generated results supporting our initial hypothesis. Thalamic bursts powerfully activate cortical neurons (*Borden et al., 2022*; *Nestvogel and McCormick, 2022*; *Ramcharan et al., 2005*; *Sherman, 1996*) via synaptic activity that can be captured by the inferred CSD. We observed a pronounced current sink near the border of layers 2/3 and 5 in MOs (near the deep stimulation site) that coincided with the timing of the thalamic rebound burst (~180 ms). This was consistent across subjects (N=9, Wilcoxon signed-rank test, p<0.05; *Figure 6A*). We also analyzed the correlation between the ERP and laminar CSD on a trial-by-trial basis and found that the EEG

signals correlated more strongly with CSD of deep compared to superficial layers (*Figure 6—figure supplement 1*). To further investigate the relationship between thalamic bursting, cortical spiking, and the ERP, we computed the correlation between these metrics in the rebound window (75–300 ms after stimulus onset) on a trial-by-trial basis for each subject. The fraction of bursting SM-TH neurons was correlated with the cortical population firing rate (8/13 mice with a significant correlation [p<0.05], mean Pearson r value: 0.4±0.04; *Figure 6B*). The cortical population firing rate was likewise correlated with the magnitude of the second, late component in the ERP (9/13 mice with a significant correlation [p<0.05], mean Pearson r value: 0.5±0.04; *Figure 6C*). Additionally, the late component in the ERP was significantly correlated with the fraction of bursting SM-TH neurons (13/13 mice with a significant correlation [p<0.05], mean Pearson r value: 0.5±0.07; *Figure 6—figure supplement 2A*), much more so than it was correlated with the SM-TH population firing rate (*Figure 6—figure supplement 2B*). These important observations link micro-scale dynamics (cellular thalamic bursting) with a macro-scale read-out (cortical EEG).

## Behavioral states modulate the ERP and the cortico-thalamic interactions

To study differences between the evoked responses across conscious and unconscious states, we performed experiments in mice that were awake and subsequently anesthetized. At the start of an experiment, we delivered up to 120 single electrical pulses while the mouse was awake, free to rest or run on a freely moving wheel. Next, we induced anesthesia with isoflurane via inhalation. Once the mouse reached a stable level of unconsciousness (no reaction to an alcohol swab placed in front of the nose; 4.0±0.3 min after induction onset at 5% isoflurane concentration), we delivered the same set of electrical stimuli.

We separated each trial by behavioral state based on whether the animal was stationary (*quiet wakefulness*), running (*active wakefulness*, defined here as an average running speed exceeding 0 cm/s in a window from –0.5 to 0.5 s from the stimulation onset), or anesthetized. We then compared average responses from 97 (median, [86–104 IQR]) quiet wakefulness, 21 (median, [16–27 IQR]) active wakefulness, and 120 (median, [120–120 IQR]) anesthetized trials across 17 animals. Both locomotion and anesthesia visibly modulated the ERPs (*Figure 7A*) and the evoked firing rates of cortical and thalamic neurons (*Figure 7B*). The second component in the ERP was diminished when running and nonexistent during anesthesia; this was captured by the decrease in ERP duration (quiet wakefulness 0.5±0.1 s, running 0.3±0.0 s, anesthetized 0.3±0.0 s; Friedman test, state effect: Q(2)=15.6, p=4.002E-4; *Figure 7C*) and the decrease in ERP magnitude (quiet wakefulness 6.1±0.9, running 1.6±0.3, anesthetized 1.6±0.3; Friedman test, state effect: Q(2)=17.4, p=1.656E-4; *Figure 7D*).

Next, we examined how cortical and thalamic activity differed as a function of behavioral state. In both regions, the baseline firing rates (measured during the pre-stimulus epoch) were higher during running and lower during anesthesia compared to quiet wakefulness (stimulated cortex baseline firing rate: N=16, quiet wakefulness 3.5±0.2 Hz, running 5.2±0.5 Hz, anesthetized 1.1±0.2 Hz; one-way repeated measures [RM] ANOVA, state effect: F[2, 30]=66.3, p=2.901E-9, *Figure 7E* left; SM-TH baseline firing rate: N=13, quiet wakefulness 9.0±1.4 Hz, running 16.4±1.5 Hz, anesthetized 0.8±0.4 Hz; Friedman test, state effect: Q[2]=24.2, p=5.689E-6, *Figure 7E* right).

The median first spike latencies for cortical and thalamic populations were the same when looking at the initial excitation (N=13; stimulated cortex latency: quiet wakefulness 4.5±0.3 ms, running 4.0±0.2 ms, anesthetized 4.6±0.4 ms; SM-TH latency: quiet wakefulness 11.5±0.8 ms, running 11.3±0.8 ms, anesthetized 10.9±0.6 ms; two-way RM ANOVA, state effect: F[2, 24]=0.3, p=0.6281, region effect: F[1, 12]=89.3, p=6.581E-7, interaction: F[2, 24]=0.5, p=0.5062; *Figure 7F*). However, the rebound spike latencies for cortex and thalamus were earlier during running compared to quiet wakefulness (N=13; stimulated cortex latency: quiet wakefulness 186.7±4.6 ms, running 163.8±5.0 ms, anesthetized 200.2±7.5 ms; SM-TH latency: quiet wakefulness 156.0±7.1 ms, running 123.1±3.9 ms, anesthetized 195.1±9.5 ms; two-way RM ANOVA, state effect: F[2, 24]=31.4, p=2.020E-7, region effect: F[1, 12]=17.2, p=1.349E-3, interaction: F[2, 24]=6.6, p=0.053; *Figure 7G*). The cortex and thalamus did not exhibit consistently timed rebound excitation during anesthesia, so the relative timing during this state was random and often later than during quiet and active wakefulness. The probability of evoking thalamic bursts decreased during running compared to quiet wakefulness and was very low (zero for many subjects) during anesthesia (N=13; median probability: quiet wakefulness 0.1 [0.01–0.24

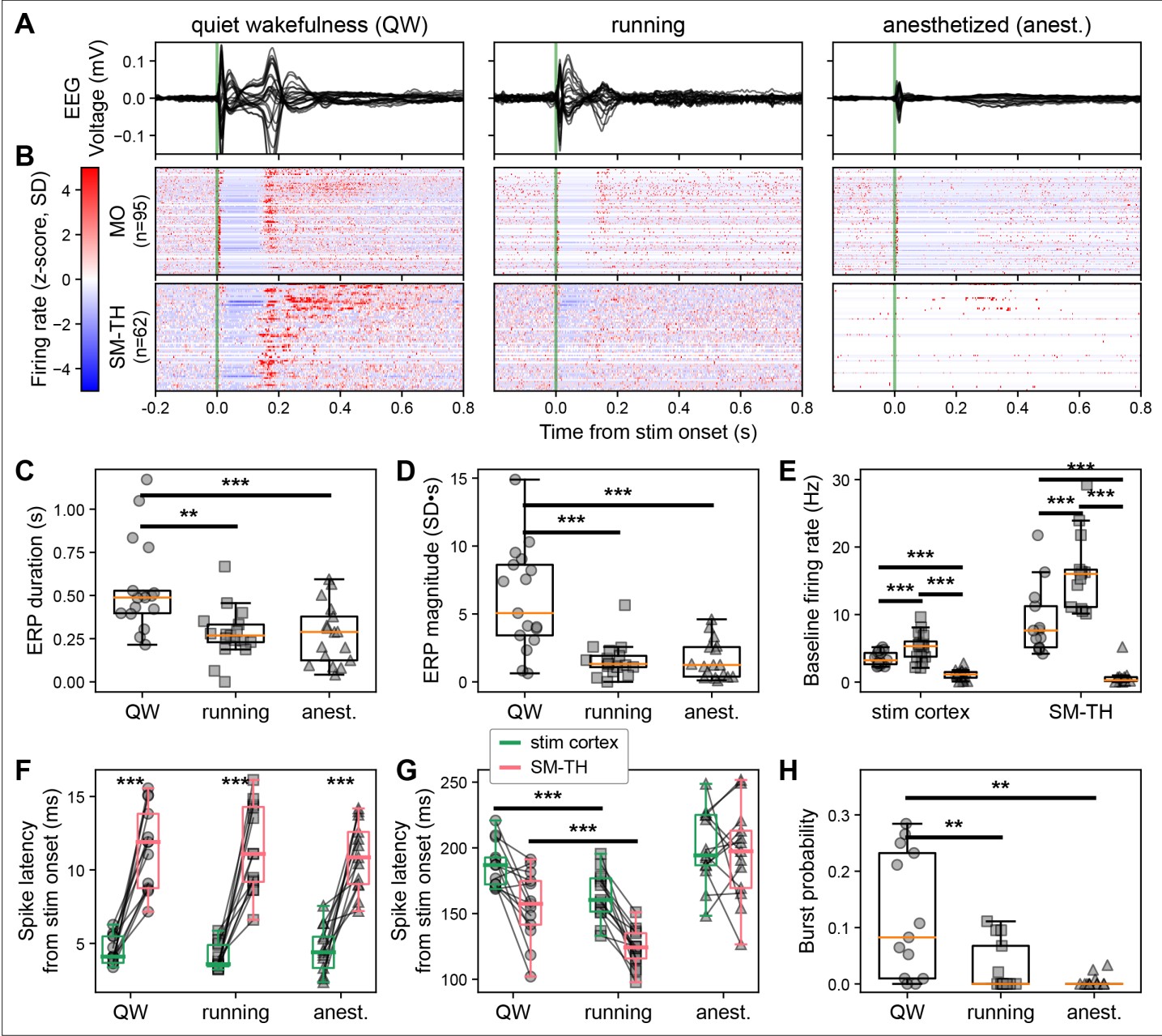

**Figure 7.** Brain state modulates the ERP via cortico-thalamo-cortical interactions. (**A**) Butterfly plot of ERPs during non-running (quiet wakefulness), running (active wakefulness), and isoflurane-anesthetized states (same subject as in *Figure 1D* bottom left). (**B**) Normalized firing rate, reported as a z-score of the average, pre-stimulus firing rate, of all RS neurons recorded by the Neuropixels probes targeting the stimulated cortex (MO) and SM-TH. (**C**) Duration and (**D**) magnitude of the ERPs for all states (see also *Figure 1E and F*): quiet wakefulness, running, and anesthetized (N=17). (**E**) Baseline rates of cortical (stim cortex) and SM-TH neurons across all states (stim cortex: N=16; SM-TH: N=13). Latency to first-spike in (**F**) early (2–25 ms) and (**G**) late (100–300 ms) windows for responsive RS neurons in the stimulated cortex (green boxes) and in the SM-TH (pink boxes) across all states. Populations were recorded simultaneously in each subject, represented by the connecting black lines (N=12). (**H**) Probability (fraction of total trials) of SM-TH spiking bursts within 75–300 ms from the stimulus onset for the three states (N=13). Boxplots show median, 25th, and 75th percentiles; whiskers extend from the box by 1.5× the IQR. One-way or two-way RM ANOVA (for normally distributed data), or Friedman test (non-parametric); * weak evidence to reject null hypothesis (0.05>p>0.01), ** strong evidence to reject null hypothesis (0.01>p>0.001), and *** very strong evidence to reject null hypothesis (0.001>p).

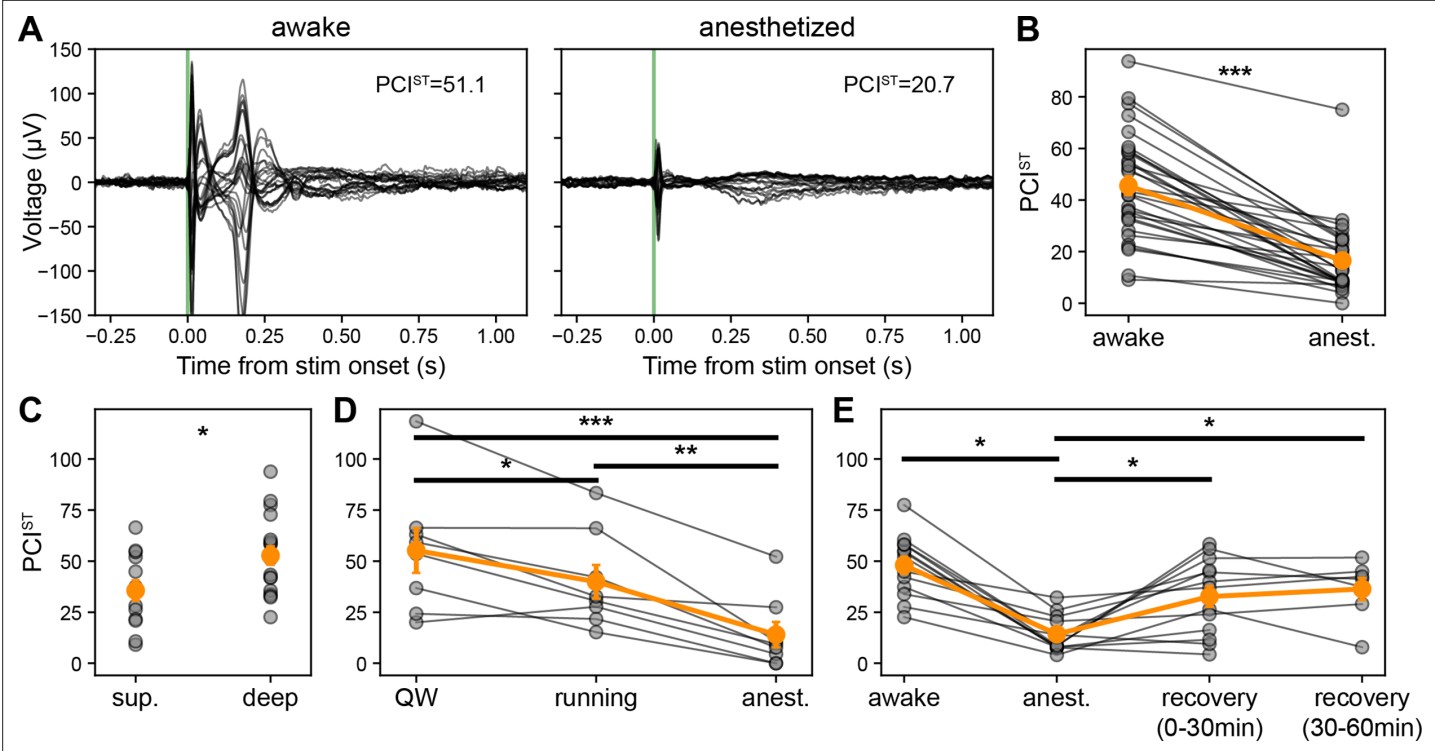

**Figure 8.** Perturbational complexity is modulated by cortico-thalamo-cortical interactions. (**A**) Butterfly plot of ERPs (−0.3 to +1.1 s) from awake (left) and anesthetized states (right). Same subject as in *Figure 1D* bottom left, annotated with PCI, state-transition (PCI$^{ST}$) values. (**B**) PCI$^{ST}$ calculated using EEG evoked responses (baseline −0.8 to −0.002 s and response 0.002–0.8 s) for awake and anesthetized states. Individual values represented with gray connected circles (N=31 sessions across 24 mice). (**C**) PCI$^{ST}$ for superficial vs. deep cortical stimulation while mice are awake (includes quiet wakefulness and running trials). (**D**) PCI$^{ST}$ for quiet wakefulness, running, and anesthetized states. Individual values represented with gray connected circles (N=8). (**E**) PCI$^{ST}$ for quiet wakefulness, anesthetized, and two subsequent recovery states. Individual values represented with gray connected circles (N=13). Orange circles and error bars represent mean ± SEM. Student's two-tailed t-test, paired t-test, or one-way RM ANOVA; * weak evidence to reject null hypothesis (0.05>p>0.01), ** strong evidence to reject null hypothesis (0.01>p>0.001), and *** very strong evidence to reject null hypothesis (0.001>p).

IQR], running 0 [0–0.07 IQR], anesthetized 0 [0–0.002 IQR]; Friedman test, state effect: Q[2]=18.6, p=9.142E-5; *Figure 7H*).

Thus, thalamic activity preceded cortical activity during the rebound excitation in both quiet and active wakefulness. However, during quiet wakefulness, the rebound was characterized by thalamic burst firing, which correlated with increased cortical spiking and higher ERP magnitudes (*Figure 6*).

## Thalamo-cortical interactions affect perturbational complexity in head-fixed mice

The differences of ERPs across behavioral states, reflecting differences in the cortico-cortical and cortico-thalamic interactions as described above, resulted in significant changes in complexity across states (*Figure 8A*). To quantify complexity, we used an algorithm that quantifies the number of state transitions for each principal component in the ERP (PCI, state-transition [PCI$^{ST}$]; *Comolatti et al., 2019*). Like the PCI, this metric distinguishes conscious from unconscious states in a variety of human volunteers and patients with disorders of consciousness (*Comolatti et al., 2019*) as well as rodents (*Arena et al., 2021*; *Cavelli et al., 2022*); unlike PCI, PCI$^{ST}$ does not require source modeling and is not upper-bounded by one. When PCI$^{ST}$ is high, signals are both spatially and temporally differentiated. When it is low, signals are highly correlated across space and time or very small relative to the pre-stimulus signal. We found that PCI$^{ST}$ was higher during the awake state than during the isoflurane-anesthetized state in all subjects (N=31; awake PCI$^{ST}$ 45.5±3.6; anesthetized PCI$^{ST}$ 16.5±2.4; paired t-test, p=2.508E-11; *Figure 8B*).

Based on our observation that superficial stimulation evoked simpler responses than deep stimulation (in the awake state), we found that PCI$^{ST}$ was, indeed, significantly lower for superficial than

for deep stimulation (superficial: N=13, awake PCI$^{ST}$ 35.6±4.8; deep: N=18, awake PCI$^{ST}$ 52.7±4.4; Student's two-tailed t-test, p=0.0179; *Figure 8C*).

To compare PCI$^{ST}$ during quiet and active wakefulness and isoflurane anesthesia, we selected subjects that had at least 30 running trials (for this comparison, trial numbers were matched within subject, on average 51 trials) since averaging the ERP over too few trials can affect the PCI$^{ST}$ calculation. *Comolatti et al., 2019* showed that PCI$^{ST}$ increased as the number of trials increased, specifically in wakefulness because averaging over more trials increases the signal-to-noise ratio of the ERP by reducing the contribution of background, non-stimulus-evoked activity. PCI$^{ST}$ was significantly different for all states; the highest complexity ERPs were seen when mice were in quiet wakefulness, slightly lower complexity ERPs during active wakefulness, with the lowest complexity during anesthesia (N=8; quiet PCI$^{ST}$ 55.2±10.3, active wakefulness 39.9±7.7; anesthetized 14.0±5.8; one-way RM ANOVA, state effect: F[2, 14]=28.3, p=1.202E-5; *Figure 8D*).

To test whether PCI$^{ST}$ recovered to baseline after anesthesia, we repeated the same set of 120 electrical stimuli up to two times over 1 hr following cessation of the isoflurane in a subset of mice. All but one subject showed an increase in PCI$^{ST}$ (reaching pre-anesthesia levels) in the first 30 min after cessation of isoflurane, but the complexity did not increase further in the subsequent 30 min (N=13; awake PCI$^{ST}$ 48.0±4.0; anesthetized PCI$^{ST}$ 14.3±2.3; recovery 0–30 min PCI$^{ST}$ 32.7±4.9; recovery 30–60 min PCI$^{ST}$ 36.4±5.0; one-way RM ANOVA, state effect: F[3, 18]=11.6, p=1.804E-4; *Figure 8E*).

## Discussion

We recorded brain-wide, multi-scale, evoked EEG, LFP, and single neuron responses to cortical electrical stimulation in head-fixed mice that were awake and, subsequently, anesthetized with isoflurane. We found that CTC interactions drive the long-lasting ERPs elicited by deep cortical stimulation during quiet wakefulness. The thalamic rebound response is characterized by thalamic burst firing that temporally coincides and is correlated with the second, late component in the ERP. Furthermore, the CTC interactions are modulated by the behavioral state of the animal, which is mirrored in the ERP.

### Electrically evoked spiking pattern depends on the depth of stimulation

Electrical stimulation of superficial layers results in a brief excitation followed by 44.3±16.4 ms of silence in local cortical neurons only (without any detected involvement of the thalamus). This pattern has been seen in different species (*Butovas et al., 2006*; *Butovas and Schwarz, 2003*; *Cavelli et al., 2022*; *Chung and Ferster, 1998*; *Contreras and Steriade, 1995*; *Douglas and Martin, 1991*; *Grenier et al., 1998*; *Hao et al., 2016*; *Kara et al., 2002*; *Logothetis et al., 2010*; *Sombeck et al., 2022*; *Vyazovskiy et al., 2013*) and is reminiscent of the bi-stability (a state characterized by spontaneous alternation between bouts of activity and periods of silence) reported in deep sleep and in unconscious patients following a brief stimulation (*Hill and Tononi, 2005*; *Pigorini et al., 2015*; *Rosanova et al., 2018*; *Timofeev et al., 2001*; *Usami et al., 2015*).

Deep electrical stimulation elicits a stereotyped triphasic spiking pattern – a brief excitation followed by 127.8±4.1 ms of silence and a *rebound* excitation – in local cortical and thalamic neurons. Other studies also describe rebound excitation at similar latencies following stimulation-evoked quiescent periods (*Butovas et al., 2006*; *Butovas and Schwarz, 2003*; *Cavelli et al., 2022*; *Grenier et al., 1998*). This local, cortical down-state may be due to a brief shift in excitatory-inhibitory balance, a large fraction of cortical cells being refractory, and/or a lack of excitatory input from the thalamus. We see similar response profiles in both RS and FS cells (*Figures 2 and 4*, *Figure 4—figure supplement 1*).

Given the direct projections from deep pyramidal neurons to thalamic regions (*Harris et al., 2019*; *Harris and Shepherd, 2015*; *Hooks et al., 2013*), it is not surprising that we observed strong recruitment of thalamic neurons 6.7 ms (on average) following the onset of cortical spiking. The thalamic spiking pattern evoked by deep stimulation during quiet wakefulness – brief excitation followed by a period of quiescence (74.6±15.5 ms) and a subsequent burst of action potentials – is consistent with previously described thalamic responses to cortical stimulation (*Contreras and Steriade, 1995*; *Grenier et al., 1998*). The fast action potential bursts in thalamic cells following the period of quiescence are likely to be triggered by low-threshold Ca$^{2+}$ spikes, known to follow periods of pronounced

hyperpolarization in thalamic relay neurons (*Borden et al., 2022*; *Contreras and Steriade, 1995*; *Grenier et al., 1998*; *Guido and Weyand, 1995*; *Halassa et al., 2011*; *Lu et al., 1992*; *Nestvogel and McCormick, 2022*; *Urbain et al., 2019*). The thalamic silence-burst pattern may be a result of a withdrawal of excitation due to the cortical down-state or disynaptic inhibition via stimulation of thalamic reticular neurons, a primary source of GABAergic input to the thalamus (*Bal et al., 2000*; *Domich et al., 1986*; *Pinault, 2004*; *Sherman and Guillery, 1996*), both leading to the low-threshold $Ca^{2+}$ spike burst. We are currently unable to distinguish between these two possible mechanisms but are planning follow-up experiments to address this important question.

Evoked thalamic bursting consistently preceded cortical rebound spiking by 37.0 ms (on average, *Figure 5F*), comparable to the latency described by *Grenier et al., 1998*. The occurrence of bursts is compatible with these being thalamic relay neurons that excite their cortical targets (*Guo et al., 2017*; *Sherman, 2001*). Numerous previous studies have shown that thalamic relay neurons exhibit *spontaneous* bursting during anesthesia (*Contreras and Steriade, 1995*; *Grenier et al., 1998*; *Swadlow and Gusev, 2001*), NREM sleep (*Domich et al., 1986*; *Halassa et al., 2011*; *Urbain et al., 2019*), drowsiness (*Stoelzel et al., 2009*; *Swadlow and Gusev, 2001*), and wakefulness (*Nestvogel and McCormick, 2022*; *Urbain et al., 2015*). In our hands, thalamic relay neurons exhibit *evoked* rebound bursting during quiet wakefulness, but not during running or isoflurane anesthesia. This is consistent with findings showing that thalamic hyperpolarization and low-threshold $Ca^{2+}$ spike-bursting coincides with movement offset in mice (*Nestvogel and McCormick, 2022*). Likewise, some anesthetics decrease the probability of thalamic bursts; specifically, isoflurane has been shown to shunt the low-threshold $Ca^{2+}$ spikes and, therefore, the associated bursting (*Ries and Puil, 1999*; this does not rule out additional cortical effects of isoflurane, e.g. *Bharioke et al., 2022*). Therefore, our finding does not contradict results from other studies reporting evoked thalamic bursting in unconscious states, such as anesthesia via pentobarbital or ketamine/xylazine (*Contreras and Steriade, 1995*; *Grenier et al., 1998*) and NREM sleep (*Urbain et al., 2019*), which are likely to affect thalamic neurons in different ways compared to isoflurane.

## Thalamo-cortical dynamics modulate perturbational complexity in mice

The behavioral state of the mouse strongly affects the underlying neural activity (*Figure 7*). Running and isoflurane anesthesia modulate the excitability of the network in opposing directions. Cortical firing rates during quiet wakefulness, 3.3±0.2 Hz, increase to 5.0±0.5 Hz for active wakefulness and decrease to 1.1±0.2 Hz during anesthesia (*Figure 7E* left); likewise in SM-TH, where baseline rates increase from 9.4±1.5 Hz during quiet wakefulness to 16.8±1.6 Hz during active wakefulness and drop to 0.8±0.4 Hz during anesthesia (*Figure 7E* right). Conversely, the median probability of electrically evoked thalamic bursting decreases from its baseline during quiet wakefulness of 9.5% ([10.0–23.7 IQR] to effectively zero for both active wakefulness [0–7.4 IQR] and anesthesia [0–0.2 IQR]). While we cannot conclusively rule out that mice were asleep during the *quiet wakefulness* periods we analyzed, we believe they were likely to be awake because the experiments were performed during the dark phase of the light/dark cycle when they are less likely to enter a sleep state (*Franken et al., 1999*), and the mice did not undergo specific training to promote drowsiness or sleep.

The stimulus-evoked bursts in the thalamus and the resulting thalamo-cortical interaction, that primarily occur when the animal is in quiet wakefulness, result in longer, higher amplitude ERPs (*Figure 7C and D*). This is consistent with previous studies showing that stimulus-evoked bursting in thalamic relay neurons robustly activates cortical neurons (*Borden et al., 2022*; *Nestvogel and McCormick, 2022*; *Ramcharan et al., 2005*; *Stoelzel et al., 2009*; *Swadlow and Gusev, 2001*), with a cortical rebound depolarization lasting up to ~0.5 s (*Grenier et al., 1998*), and 3–5 Hz cortical oscillation of ~1 s median duration (*Nestvogel and McCormick, 2022*). Indeed, the more thalamic cells burst during the rebound period, the larger the magnitude of the second, late component in the ERP (*Figure 6—figure supplement 2A*). This observed link between the ERP and activity in the CTC loop, an important finding made possible by our unique experimental design, may help interpret the complex physiological and behavioral effects of electrical or magnetic stimulation in the human brain (*Borchers et al., 2011*).

ERPs are associated with higher $PCI^{ST}$ values during awake than during isoflurane anesthesia (quiet wakefulness $PCI^{ST}$ 55.2±10.3; active wakefulness $PCI^{ST}$ 39.9±7.7; anesthetized $PCI^{ST}$ 14.0±5.8), consistent with prior work in humans (*Comolatti et al., 2019*) and in rodents (*Arena et al., 2021*; *Cavelli*

*et al., 2022*). Furthermore, the ERPs are shorter, and the associated PCI$^{ST}$ is lower during active wakefulness than during quiet wakefulness, possibly due to a lower probability of evoked thalamic bursting. All subjects have higher PCI$^{ST}$ during active wakefulness compared to anesthesia (active wakefulness PCI$^{ST}$ 39.9±7.7; anesthetized PCI$^{ST}$ 14.0±5.8). The comparison of PCI$^{ST}$ across quiet wakefulness, active wakefulness, and anesthesia suggests that in mice there is an optimal network state which maximizes complexity, reminiscent of the inverted-U relationship between arousal and task performance (*McGinley et al., 2015*; *Yerkes and Dodson, 1908*).

The last decade has witnessed the emergence of the mouse as a model organism to study the neuronal correlates of consciousness (*Aru et al., 2020*; *Bharioke et al., 2022*; *Koch et al., 2016*; *Larkum, 2013*; *Sachidhanandam et al., 2013*; *Suzuki and Larkum, 2020*). These studies point to the critical role of layer 5 pyramidal neurons, as well as cortico-cortical feedback to apical dendrites of infragranular pyramidal neurons. Although the present study was not designed with a classical perceived vs. non-perceived paradigm in mind (*Koch et al., 2016*), our finding of shorter and less complex ERPs and a reduced effectiveness of the CTC circuit during isoflurane anesthesia is certainly compatible with such results. To understand the extent to which our results generalize, it would be important to apply the PCI method to a diversity of different anesthetics in mice (e.g. *Arena et al., 2021*; *Bharioke et al., 2022*; *Cavelli et al., 2022*) and investigate the role of thalamic bursting and CTC activity in relation to large scale EEG dynamics in such states.

## Thalamic-mediated local neural responses underlie global ERP in distant cortical regions

Based on the presence of dense intercortical connections between areas (from neurons in layers 2/3–6; *Harris et al., 2019*), we expected that cortical stimulation during quiet wakefulness would directly elicit synaptic and spiking activity in other cortical sites outside the stimulated area. We observed large-amplitude evoked activity in the EEG and LFP in ipsilateral cortical regions close (0.5–1 mm) and up to 5 mm away from the stimulated site (*Figures 2 and 3*; *Buzsáki et al., 2012*; *Kajikawa and Schroeder, 2011*), consistent with complex evoked LFP responses in distal ipsilateral and contralateral cortical sites in response to electrical and optogenetic stimulation in rodents (*Cavelli et al., 2022*).

Given that the LFP reflects volume conduction and not only local activity (*Buzsáki et al., 2012*; *Kajikawa and Schroeder, 2011*), we examined CSD and population spiking of both stimulated and non-stimulated cortical regions (*Figure 3*). Deep cortical stimulation elicits measurable responses in both CSD and population spiking, whereas we were unable to detect such activations in non-stimulated cortical areas from superficial stimulation. Similarly, *Cavelli et al., 2022* showed that evoked laminar LFP responses were either undetectable or smaller when electrical stimulation was applied to the most superficial layers of the rat cortex compared to deeper layers, reflected by smaller PCI$^{ST}$ values. The Neuropixels probes were inserted approximately perpendicular to the cortical surface. Deviation from perpendicular will impact the CSD derivation by increasing the 'effective' cortical thickness. This should not significantly affect the location of sources and sinks identified by the analysis.

The effects of electrical microstimulation are thought to be predominantly mediated by activation of axons (*Gustafsson and Jankowska, 1976*; *Hao et al., 2016*; *Histed et al., 2009*; *Nowak and Bullier, 1998*; *Tehovnik et al., 2006*). Therefore, the stronger evoked responses to deep stimulation could be due to activation of layers 5 and 6 cortico-thalamic cells via direct activation of axons that primarily terminate in thalamic regions (*Figure 5*), consistent with the literature suggesting that subcortical projecting, myelinated axons are more excitable than non-myelinated horizontal fibers within the cortex (*Tehovnik and Slocum, 2013*). Because most of the modulation in non-stimulated cortical regions occurs in the late time windows (25–300 ms after stimulation onset; *Figure 4F*), and the CSD analysis shows consistent sources/sinks across subjects only for late times (*Figure 6A*), we speculate that involvement of non-stimulated cortical areas is due to bursting in common thalamic projections and underscores the role of the CTC loop in perturbational complexity.

## Limitations and benefits of studying ERPs in mice

There are qualitative differences between the ERPs we observed in mice and those in people (*Casali et al., 2013*; *Comolatti et al., 2019*; *Ferrarelli et al., 2010*; *Massimini et al., 2005*): most importantly, the two robust components (at 25 and 180 ms post-stimulation) in the ERPs from mice in quiet wakefulness are not apparent in human TMS-evoked EEG. Of course, there are several key differences

between the brains of these two species, the most important being the three orders of magnitude difference in volume and number of neurons. The mouse brain is roughly 0.5 cm³ and contains 71 million neurons, whereas the human brain is closer to 1200 cm³ with 86 billion neurons (*Herculano-Houzel et al., 2006*; *von Bartheld et al., 2016*; *Walløe et al., 2014*). The smooth mouse neocortex is 0.8–1 mm thick, with 14 million neurons, whereas the highly folded human neocortex is 2.5–3.0 mm thick with 16 billion neurons (*Herculano-Houzel et al., 2006*; *Walløe et al., 2014*). Recently, *Bakken et al., 2021* published a comprehensive comparative analysis of cell types across human, marmoset, and mouse cortex. A key difference they highlight is more layer 2/3 intratelencephalic neurons in primates compared to mice, whereas layer 5 extratelencephalic and layer 6 corticothalamic neurons were significantly more common in mice. This is consistent with our findings that show pronounced cortico-thalamic activation and little evidence of wide-spread cortico-cortical activation in mice.

Another possible explanation for the striking difference in the ERPs between the two species is the different biophysical modes of stimulation – direct current flow from and to the inserted electrode vs. a magnetic-field induced current flow in the cortical tissue underneath the TMS coil resting against the scalp. Electrical stimulation directly activates axons that run near the stimulation site, which can lead to antidromic activation of cell bodies and/or orthodromic propagation to the axon targets (*Borchers et al., 2011*; *Gustafsson and Jankowska, 1976*; *Histed et al., 2009*; *Nowak and Bullier, 1998*; *Sombeck et al., 2022*; *Tehovnik et al., 2006*; *Tehovnik and Slocum, 2013*; *Terao and Ugawa, 2002*), whereas TMS elicits indirect waves of cortical activation by exciting axons and neurons mainly trans-synaptically (*Pashut et al., 2014*; *Siebner et al., 2022*; *Terao and Ugawa, 2002*). However, there are also many similarities – stimulation in awake humans evokes a suppression of high frequencies, presumed to be an *off* period, close to the stimulated site (*Pigorini et al., 2015*). This is in line with the down-state we observed in mice, which highlights the role of the perturbational technique used for complexity measurements.

Despite these differences, this study led to novel and unexpected results linking the ERP to activity in the CTC loop. There are many additional open questions regarding the neural mechanisms underlying spontaneous and stimulus-evoked EEG signals (*Cohen, 2017*). Rodent models offer a unique opportunity to combine EEG recordings with high-density extracellular recording technology to shine light on the underlying microcircuit dynamics. This study opens the door for future work with rodents using different perturbational techniques (e.g. chemogenetics, optogenetics [*Cavelli et al., 2022*], and TMS [*Senda et al., 2021*]) to causally link the contributions of different cell types, brain regions, and network dynamics to EEG signals and ERP features commonly used in clinical and research settings.

## Methods

Experimental procedures closely followed those described in *Siegle et al., 2021*. A summary of these methods and details of procedures that differ are provided below.

### Mice

Mice were maintained in the Allen Institute animal facility and used in accordance with protocols approved by the Allen Institute's Institutional Animal Care and Use Committee under protocols 1703 and 2003. All experiments used C57BL/6J wild-type mice (N=37). Male and female wild-type C57BL/6J mice were purchased from Jackson Laboratories (JAX stock #000664) at postnatal day 28, and they were 9–28 weeks old at the time of all in vivo electrophysiological recordings.

After surgery, all mice were single-housed and maintained on a reverse 12 hr light cycle in a shared facility with room temperatures between 20 and 22°C and humidity between 30 and 70%. All experiments were performed during the dark cycle. Mice had ad libitum access to food and water.

### Surgical procedures and habituation

Each mouse went through the following order of procedures prior to the day of the experiment: (1) an initial sterile surgery to implant an EEG array and a titanium headframe for head-fixed electrophysiological experiments in vivo; (2) 5 days of recovery time post-surgery; (3) at least 3 weeks of habituation to head-fixation; (4) a second sterile surgery to perform small craniotomies to allow for insertion of the stimulating electrode and Neuropixels probes.

1–3 hr prior to each surgery, pre-operative injections of dexamethasone (3–4 mg/kg, IM), and ceftriaxone (100–125 mg/kg, SC) were administered. Mice were deeply anesthetized with isoflurane (5% isoflurane induction and 1.5–2.5% maintenance) and placed in a stereotaxic frame. Vital signs were monitored, body temperature was maintained at 37.5°C with a heating pad under the animal (TC-1000 temperature controller, CWE, Inc), and ocular lubricant (I Drop, VetPLUS) was applied to maintain hydration of the eyes during anesthesia. Atropine (0.02–0.05 mg/kg, SC) and carprofen (5–10 mg/kg, SC) were administered at the start of the procedure. After the surgical procedure, mice received an injection of lactated Ringer's solution (up to 1 mL, SC) and recovered on a heating pad. Animals received 2 days of analgesics and antibiotics post-surgery.

The initial surgery was performed on healthy mice that ranged in age from 5 to 20 weeks. Mice were deeply anesthetized prior to removing skin and exposing the skull. After leveling the skull, a 30-electrode mouse EEG array (NeuroNexus Technologies, Inc, Ann Arbor, MI, USA) was carefully positioned on top of the skull and fixed in place with Kwik-Cast (World Precision Instruments, Inc, Sarasota, Florida). Skull screws were implanted either over the right olfactory bulb or left and right cerebellum that functioned as reference and ground for the EEG signals. White C&B Metabond (Parkell, Inc, Edgewood, NY, USA) was then used to secure a custom titanium headframe and the EEG array to the skull. After 5 days of recovery, mice spent at least 3 weeks being habituated to handling and head-fixation.

Following habituation and up to 1 day before the recording, mice underwent the second surgical procedure. Under a microscope, the skull was exposed by drilling through the outer layer of Metabond then removing the Kwik-Cast and a small portion of the EEG polymer substrate, carefully avoiding damage to the EEG contacts. Up to three small craniotomies (less than 0.5 mm in diameter) were drilled to allow access to the brain regions of interest for the subsequent experiment. A small piece of artificial cerebrospinal fluid (ACSF)-soaked gel-foam sponge was positioned on top of each craniotomy and Kwik-Cast was used to seal it. A 3D-printed plastic well was also fixed to the existing Metabond around the craniotomies.

## *In-vivo* recording and stimulation

The day of the experiment, the mouse was placed on the running wheel and fixed to the headframe clamp with two set screws. Next, the thin layer of Kwik-Cast was removed to expose the craniotomies, and abundant ACSF was added on top of the skull to prevent the exposed brain tissue from drying out. A 3D-printed cone was lowered to prevent the mouse's tail from contacting the probes, and a black panel was placed over the front of the rig, placing the mouse in complete darkness.

## EEG recording

All 30 EEG electrodes were connected to a 32-channel head-stage (RHD 32ch, Intan Technologies, Los Angeles, CA, USA) controlled by an Open Ephys acquisition board (*Siegle et al., 2017*). EEG signals were sampled at 2.5 kHz with a low-frequency cutoff of 0.1 Hz and digitized with 16-bit resolution. Each circular, platinum electrode on the EEG array had a diameter of 500 μm and an impedance of 0.01 MΩ (impedance measured by the manufacturer). All subjects had a skull screw implanted over the right cerebellum (penetrating the skull but not the dura) that served as electrical ground. One of the following served as a common reference electrode for the EEG signals: a 0.7×0.7 mm square platinum surface electrode above the left cerebellum, a skull screw implanted over the right olfactory bulb, or a skull screw implanted over the left cerebellum.

## Neuropixels recording

In a subset of mice (N=26/37), simultaneous recordings from the EEG array and multiple Neuropixels probes (*Jun et al., 2017*) were performed. These experiments used Neuropixels 3a prototypes or standard Neuropixels 1.0 probes configured to record from the 384 electrodes closest to the tip of the probe. The signals from each recording site were split in hardware into a spike band (30 kHz sampling rate, 500 Hz high-pass filter, 500× gain) and an LFP band (2.5 kHz sampling rate, 1000 Hz low-pass filter, 250× gain), and data was acquired using the Open Ephys GUI (*Siegle et al., 2017*). Each probe was either connected to a dedicated field-programmable gate array streaming data over Ethernet (Neuropixels 3a) or a PXIe card inside a National Instruments chassis (Neuropixels 1.0).

The reference connection on the Neuropixels probes was permanently soldered to ground using a silver wire, and all recordings were made using either an external reference configuration (Neuropixels 3a) or a tip reference configuration (Neuropixels 1.0). All Neuropixels head-stage grounds, which were contiguous with the probe grounds, were connected in parallel to animal ground – a 32 AWG silver wire (A-M Systems, Sequim, Washington) placed in the ACSF on top of the craniotomies.

### Neuropixels insertion

During the acute electrophysiological experiment, up to three Neuropixels probes were inserted targeting motor (MO), anterior cingulate (ACA), somatosensory (SS), visual (VIS), and thalamic nuclei. We restricted our analysis of thalamic nuclei to those that had strong projections to and/or from motor and somatosensory areas (*Guo et al., 2017*; *Harris et al., 2019*): anteroventral (AV), central lateral (CL), mediodorsal (MD), posterior (PO), reticular (RT), ventral anterior-lateral (VAL), ventral posterolateral (VPL), ventral posteromedial (VPM), and ventral medial (VM). We refer to these collectively as the somatomotor-related thalamic nuclei (SM-TH), disregarding the distinction between first and higher order thalamic regions.

The probe insertion process followed the steps detailed by *Siegle et al., 2021*. Briefly, each probe was individually inserted into the brain using a three-axis micromanipulator (New Scale Technologies, Victor, NY, USA) at a rate of 200 µm per min to a depth of 3.5 mm or less in the brain. After the probes reached their targets, they were allowed to settle for 10–15 min before starting the experiment. Before insertion, all Neuropixels probes were coated with a fluorescent dye (Vybrant DiI/DiO/DiD, ThermoFisher Scientific, Waltham, MA, USA) by repeatedly immersing them in a well filled with the dye and removing each probe slowly allowing the dye to dry on the surface.

### Behavioral data and synchronization

The angular position of the running wheel and the synchronization signal for the electrical stimulus were acquired by a dedicated computer with a National Instruments card acquiring digital inputs at 100 kHz, which was considered the master clock. A 32-bit digital 'barcode' was sent with an Arduino Uno (DEV-11021, SparkFun Electronics, Niwot, CO, USA) every 30 s to synchronize all devices with the neural data from the EEG array and the Neuropixels probes. Details regarding the post-hoc data synchronization using the barcodes are described by *Siegle et al., 2021*.

### Cortical stimulation

Electrical stimulation was delivered through a custom bipolar platinum-iridium stereotrode (Microprobes for Life Science, Gaithersburg, MD, USA) consisting of two parallel monopolar electrodes (50 kΩ impedance) with a vertical offset of 300 µm between the two tips. During each stage of the experiment, up to 120 biphasic, charge-balanced, cathodic-first current pulses (200 µs per phase, 3.5–4.5 s inter-stimulus interval) were delivered at three different current intensities (360 pulses total). The current intensities were chosen for each animal while it was awake before starting the experiment based on the following criteria: (1) the electrical pulse did not evoke any visible twitches and (2) the medium current elicited visible averaged evoked responses for most of the EEG electrodes (n>15) defined as three SDs above baseline for at least 100 ms following the stimulus onset.

The stimulation electrode was acutely inserted using a three-axis micromanipulator, like the Neuropixels probes. We targeted four locations for stimulation: MOs, layer 2/3 (N=10 mice, 0.37±0.04 mm below the brain surface); MOs, layer 5/6 (N=11 mice, 1.15±0.06 mm below the brain surface); SSp, layer 2/3 (N=8 mice, 0.47±0.06 mm below the brain surface); and SSp, layer 5/6 (N=11 mice, 0.92±0.05 mm below the brain surface). Some mice received stimulation in multiple locations (N=10). Before insertion, the stimulation electrode was coated with a fluorescent dye, like the Neuropixels probes.

### Experimental timeline

The paired EEG/Neuropixels recordings were acute, lasting 2–3 hr. At the beginning, awake head-fixed mice (free to run on or rest on the running wheel) were exposed to electrical stimuli after 5 min of baseline recording without stimulation. After the awake session, mice were anesthetized via inhaled isoflurane (5% induction) delivered through a small tube placed in front of the mouse's nose. Once a surgical level of anesthesia was reached, the animal was maintained unconscious by keeping the

isoflurane level at 1–1.5%. The unconscious state was verified by lack of reaction to noxious stimuli such as toe or tail pinch and alcohol pad close to the nose of the animal. The EEG signals were also continuously monitored to avoid the burst suppression mode indicative of very deep anesthesia (*Purdon et al., 2015*). Then the same electrical stimuli session was repeated. While under anesthesia, we recorded the precise concentration of isoflurane delivered over time, in addition to the data streams listed above. Finally, the isoflurane was turned off and the mouse recovered. In a subset of mice (N=16), the electrical stimuli were delivered for a third (and sometimes a fourth) time during the recovery period. The resulting dataset allowed a direct comparison of the evoked neural responses across different brain states (quiet wakefulness, running, anesthetized, recovery).

## Probe removal and cleaning
Upon completion of the experiment, probes were retracted from the brain at a rate of 1 mm/s. The craniotomies were protected with a small piece of ACSF-soaked gel-foam sponge and a thin layer of Kwik-Cast before mice were removed from head fixation and returned to their home cages overnight. The probes and the stimulating electrode were immersed in a well of 1% Tergazyme for around 2–6 hr to remove residual tissue, and then rinsed in purified water.

## Quality control for the EEG experiments
### Overall EEG signal quality
Before the experiment, the EEG signals were tested by exposing the animal to visual flashes and evaluating the signal-to-noise ratio of the EEG evoked responses. Animals with low signal-to-noise ratio, high levels of 60 Hz noise, or large movement artifacts were not used for the experiment (8 mice).

### EEG stimulation artifact
If more than half of the EEG channels over the course of the experiment showed large stimulation artifacts lasting for hundreds of milliseconds after the stimulation onset, the session was excluded (8 sessions across 6 mice). In total, out of 37 implanted mice, 23 were included in the study.

## Ex vivo imaging and localization of electrodes
After the experiment, mice were deeply anesthetized (5% isoflurane) and perfused with 4% paraformaldehyde. The brains were preserved in 4% paraformaldehyde for 48 hr, rinsed with 1× PBS, and stored at 4°C in PBS. The brains were then processed in one of two ways: brains were sliced into 100 μm coronal sections using a vibratome (Leica VT1000S) and imaged with a fluorescent microscope (Olympus VS110/120) at 10× magnification, or whole brains were imaged using serial two-photon tomography (*Oh et al., 2014*; *Ragan et al., 2012*).

Images of the 100 μm coronal sections were aligned to the Allen Institute Common Coordinate Framework (CCFv3) following the process detailed by *McBride et al., 2023* and images from serial two-photon tomography were aligned to the CCFv3 following the process detailed by *Oh et al., 2014*. Fluorescent tracks corresponding to the location of the Neuropixels probes and the stimulation electrode were manually identified in the aligned images. Because each CCFv3 coordinate corresponds to a unique brain region, the precision of brain region assignments is determined by the resolution of the CCFv3, which was 25 μm per pixel. For each Neuropixels probe, the locations of major structural boundaries along the track was manually aligned with the physiology data (*Liu et al., 2021*; *Siegle et al., 2021*). Then each recording channel along the Neuropixels probe (and associated neurons) was assigned to a unique CCFv3 structure. Other studies have reported better than 0.1 mm accuracy for electrode localization following similar methods (*Liu et al., 2021*).

## Data processing
### Stimulus artifact masking
In all recordings, we observed short latency voltage transients time-locked to the stimulus. Though the stimulus had a total duration of 0.4 ms, artifacts sometimes lasted up to 2 ms. To mask the artifact, the raw signal from –2 to 0 ms was copied, reversed, and replaced the 0 to 2 ms artifact window. The same masking procedure was used across all electrophysiology data types.

## EEG

After artifact masking, EEG recordings were visually inspected to identify electrodes containing noise artifacts or remaining large and long stimulation artifacts. These were excluded from further analysis, leaving an average of 23±4 artifact-free electrodes out of 30 for each subject. EEG signals from all good electrodes were re-referenced to the common average across electrodes and bandpass filtered from 0.1 to 100 Hz (Butterworth filter, third order). Finally, the continuous EEG signals were segmented into epochs from –2 to +2 s from stimulus onset and saved for further analysis.

## Neuropixels LFP

After artifact masking, LFP signals were down sampled from 2.5 kHz to 1.25 kHz after applying an anti-aliasing filter. The signals were then high pass filtered (Butterworth filter, first order) and re-referenced to the median signal from electrodes that were in the ACSF above the brain surface (to remove common signal from the tip reference electrode). Finally, the continuous LFP signals were segmented into epochs from –2 to +2 s from stimulus onset and saved for further analysis.

## Neuropixels CSD

The preprocessed LFP epochs underwent an automatic channel rejection based on Chebyshev's inequality, iteratively interpolating any channel whose amplitude instantaneously exceeded ±7 SDs with respect to the others (*Russo et al., 2021*). The cleaned LFP voltages were smoothed in the time (window = 8 ms) and space domain (first smoothing window = 26 channels; second smoothing window = 4 channels). The CSD was calculated as the second spatial derivative (*Mitzdorf, 1985*) from the cleaned, smoothed LFP signals. The CSD formulation employed assumes an ohmic conductive medium, constant extracellular conductivity ($\sigma$=0.3 S/m), and homogeneous in-plane neuronal activity, with the boundary condition of zero current outside the sampled area.

## Spike sorting

After applying the artifact masking to the raw spike band data, it was pre-processed and spike-sorted using Kilosort 2.0 (*Stringer et al., 2019*) as described by *Siegle et al., 2021*. After spike sorting, any spikes that occurred during the artifact window (0 to +2 ms from stimulus onset) were removed from further analysis. High quality units were identified for further analysis using metrics described by *Siegle et al., 2021*. We classified RS and FS neurons (putative pyramidal and inhibitory neurons, respectively) based on their spike waveform duration (RS duration >0.4 ms; FS duration ≤0.4 ms *Barthó et al., 2004*; *Bortone et al., 2014*; *Bruno and Simons, 2002*; *Jia et al., 2019*; *Niell and Stryker, 2008*; *Sirota et al., 2008*).

## Data analysis

### EEG event-related potentials (ERPs)

Across each experiment, trials were classified by the behavioral state of the animal: quiet wakefulness, if the mouse's speed (measured by the wheel's angular velocity) was equal to 0 cm/s from –0.5 to +0.5 s from the stimulus onset; running, if the mouse's speed was greater than 0 cm/s; anesthetized, when the mouse was unconscious; and recovery, after the isoflurane delivery was discontinued. ERPs associated with a given state were then compiled by averaging all EEG traces assigned to that state.

### ERP magnitude and duration

To quantify the ERP metrics, the trial-averaged EEG signals (–2 to +2 s after stimulus onset) were used to calculate the global field power (global mean field power) by taking the SD across all channels for every time point (*Cohen, 2014*; *Esser et al., 2006*; *Lehmann and Skrandies, 1980*). Next, the z-score of the global field power was computed relative to baseline values (−2–0 s). The duration of the response was measured as the length of time the global field power (z-score) was greater than 3 SDs above baseline (z=3) in the response window (0 to +2 s). The magnitude of the response was calculated by integrating under the global field power (z-score) but above z=3 in the response window and dividing it by the area under the baseline window (see *Figure 1—figure supplement 2*).

## LFP, CSD, and population spiking magnitude

To quantify the magnitude of the evoked LFP and CSD response, the spatial average of the rectified, trial averaged signals for all channels in the respective cortical region, all layers except for layer 1. Then the AUC was calculated for the post-stimulus period (0 to +0.5 s) of the grand average (z-score of the baseline). A similar procedure was used for the population spiking, except the signals used were single neuron spike density functions. For all data types, the magnitude of the background activity was determined using a permutation test to shuffle the event onset time per trial (n=1000 shuffles), thus creating a distribution of null values representing the AUC of the background activity. The response magnitude was then calculated by subtracting the mean of the onset-shuffled values and dividing by the SD.

## Significantly modulated neurons

The fraction of neurons modulated by the electrical stimulation was quantified following the method detailed by *McBride et al., 2023*. Briefly, for each neuron, pre- and post-stimulus spikes were counted on a trial-by-trial basis for the specified windows: 2–25 ms, 25–150 ms, and 150–300 ms. The duration of the window was always matched for pre- and post-stimulus. Then a Wilcoxon signed-rank test was performed on the trial-wise pre- and post-stimulus spike counts to obtain a p-value. The p-values obtained for all RS neurons in all areas of interest were corrected using the Benjamini-Hochberg correction with the false discovery rate set at 0.05. Neurons with an adjusted p-value<0.05 were considered significantly modulated. They were labeled as 'increased' or 'decreased' based on whether their spike count increased or decreased, respectively, in the post-stimulus window relative to the pre-stimulus window on average.

## Population CSD

To compare CSD across mice, we transformed each CSD trace to a homogeneous space (10 bins per layer) through a non-linear transform. Significantly consistent activations were assessed through a Wilcoxon signed-rank test (threshold p<0.05). Isolated significant bins (i.e. not surrounded by significant bins in both space and time) were removed.

## PCI, state-transition

$PCI^{ST}$ was used to evaluate the spatiotemporal complexity of the EEG ERPs for each brain state (*Comolatti et al., 2019*). $PCI^{ST}$ was computed using the code available at https://github.com/renzocom/PCIst (*Comolatti, 2021*), and the parameters were set as follows: baseline window = (–0.8, –0.002) s; response window = (0.002, 0.8) s; minimum SNR = 1.6; maximum variance = 99%; and the parameter k=1.2. In all calculations, the number of trials per state within subject was matched.

## Statistics

Statistical analysis was performed in Python. All data was tested for normality using the Shapiro-Wilk test from the SciPy package (scipy.stats.shapiro). For normally distributed datasets, we used the unpaired t-test (scipy.stats.ttest_ind) or the paired t-test (scipy.stats.ttest_rel). For non-parametric datasets, we used the Mann-Whitney U test (scipy.stats.mannwhitneyu) or the Wilcoxon signed-rank test (scipy.stats.wilcoxon). All comparisons using t-tests are two-sided. Statistical comparisons between multiple groups were performed using both parametric (ANOVA) and non-parametric (Friedman) tests with post-hoc, pairwise comparisons corrected for multiple comparisons using Benjamini-Hochberg false discovery rate. Correlational analyses were performed using Pearson correlation coefficients.

In all figures, the convention is * weak evidence to reject null hypothesis (0.05>p>0.01), ** strong evidence to reject null hypothesis (0.01>p>0.001), and *** very strong evidence to reject null hypothesis (0.001>p). The values in the text indicate the mean and SEM (mean ± SEM) or the median and IQR (25th and 75th percentiles) for normally distributed and non-parametric datasets, respectively.

## Acknowledgements

We thank Anton Arkhipov, Matias Cavelli, Chiara Cirelli, Graham Findlay, Saurabh Gandhi, Soo Yeun Lee, Rong Mao, Marcello Massimini, Dana Mastrovito, Ethan McBride, David McCormick, Stefan Mihalas, Andrea Pigorini, Simone Sarasso, and Giulio Tononi for helpful discussions and intellectual

contributions. We thank the Animal Care and Lab Animal Services teams for mouse husbandry and care and the Manufacturing and Process Engineering team for experimental hardware and software support. We gratefully acknowledge funding from the Tiny Blue Dot Foundation (Santa Monica, California) for this study. We thank the Allen Institute founder, Paul G Allen, for his vision, encouragement, and support.

## Additional information

### Competing interests

Christof Koch: CK is a Board Member and has a financial interest in Intrinsic Powers Inc. The other authors declare that no competing interests exist.

### Funding

| Funder | Grant reference number | Author |
|---|---|---|
| Tiny Blue Dot Foundation | | Christof Koch |

The funders had no role in study design, data collection and interpretation, or the decision to submit the work for publication.

### Author contributions

Leslie D Claar, Conceptualization, Data curation, Formal analysis, Investigation, Visualization, Methodology, Writing – original draft, Writing – review and editing; Irene Rembado, Conceptualization, Data curation, Formal analysis, Investigation, Methodology, Writing – original draft, Writing – review and editing; Jacqulyn R Kuyat, Lydia C Marks, Data curation, Investigation, Writing – review and editing; Simone Russo, Data curation, Formal analysis, Writing – review and editing; Shawn R Olsen, Conceptualization, Supervision, Methodology, Writing – review and editing; Christof Koch, Conceptualization, Supervision, Funding acquisition, Methodology, Writing – original draft, Writing – review and editing

### Author ORCIDs

Leslie D Claar ![orcid] http://orcid.org/0000-0002-4030-9171
Simone Russo ![orcid] http://orcid.org/0000-0002-7762-1900
Shawn R Olsen ![orcid] http://orcid.org/0000-0002-9568-7057
Christof Koch ![orcid] http://orcid.org/0000-0001-6482-8067

### Ethics

All animal procedures were approved by the Institutional Animal Care and Use Committee at the Allen Institute under protocols 1703 and 2003.

Reviewer #1 (Public Review): https://doi.org/10.7554/eLife.84630.3.sa1
Reviewer #2 (Public Review): https://doi.org/10.7554/eLife.84630.3.sa2
Author response https://doi.org/10.7554/eLife.84630.3.sa3

## Additional files

### Supplementary files

• MDAR checklist

### Data availability

The data generated and analyzed in this manuscript are publicly available in Neurodata Without Borders (NWB) format (*Rübel et al., 2022*) in the DANDI Archive.

The following dataset was generated:

| Author(s) | Year | Dataset title | Dataset URL | Database and Identifier |
|---|---|---|---|---|
| Claar LD, Rembado I, Kuyat JK, Russo S, Marks LC, Olsen SR, Koch C | 2023 | Simultaneous electroencephalography, extracellular electrophysiology, and cortical electrical stimulation in head-fixed mice | https://dandiarchive. org/dandiset/000458 | DANDI Archive, DANDI:000458/0.230317.0039 |

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
