## [Editor Report · eLife assessment]

This study makes a **fundamental** observation about the role of activity in the mouse thalamus on scalp recorded voltage fluctuations. The novel approach and sophisticated analysis of neural signals provides **compelling** support for the authors' observations. This work will likely be of broad interest to neuroscientists.

---

## [Referee Report · Reviewer #1 (Public Review)]

The authors investigated state-dependent changes in evoked brain activity, using electrical stimulation combined with multisite neural activity across wakefulness and anesthesia. The approach is novel, and the results are compelling. The study benefits from in depth sophisticated analysis of neural signals. The effects of behavioral state on brain responses to stimulation are generally convincing.

It is possible that the authors' use of "an average reference montage that removed signals common to all EEG electrodes" could also remove useful components of the signal, which are common across EEG electrodes, especially during deep anesthesia. For example, it is possible (in fact from my experience I would be surprised if it is not the case) that under isoflurane anesthesia, electrical stimulation induces a generalized slow wave or a burst of activity across the brain. Subtracting the average signal will simply remove that from all channels. This does not only result in signals under anesthesia being affected more by the referencing procedure than during waking, but also will have different effects on different channels, e.g. depending on how strong the response is in a specific channel.

---

## [Referee Report · Reviewer #2 (Public Review)]

This study reports a novel role of thalamic activity in the late components of a cortical event related potential (ERP). To show this association, the authors used high-density EEG together with multiple deep electrophysiological recordings combined with electrical stimulation of superficial and deep cortical layers. Stimulation of deep layers elicits a late ERP component that is closely related to bursts of thalamic activity during quiet wakefulness. This relationship is quite noticeable when deep layers of the cortex are stimulated, and it does depend on arousal state, being maximal during quiet wakefulness, diminished during active wakefulness, and absent during anesthesia.

The study is very well performed, with a high number of subjects and appropriate methodology. Performing simultaneous recording of EEG and several neuropixels probes together with cortical microstimulation is no small feat considering the size of the mouse head and the fact that mice are freely behaving in many of the experiments. It is also noticeable how the authors use a seemingly outdated technique (electrical microstimulation) to produce compelling and significant research. The conclusions regarding the thalamic contributions to the ERP components are strongly supported by the data.

The spatiotemporal complexity is almost a side point compared to what seems to me the most important point of the paper: showing the contribution of thalamic activity to some components of the cortical ERP. Scalp ERP's have long been regarded as purely cortical phenomena, just like most of EEG, and this study shows convincing evidence to the contrary.

The data presented seemingly contradicts the results presented in Histed et al. (2009), who asserts that cortical microstimulation only affects passing fibers near the tip of the electrodes, and results in distant, sparse, and somewhat random neural activation. In this study, it is clear that the maximum effect happens near the electrodes, decays with distance, and it is not sparse at all, suggesting that not only passing fibers are activated but that also neuronal elements might be activated by antidromic propagation from the axonal hillock. This appears to offer proof that microstimulation might be much more effective than it was thought after the publication of Histed 2009, as the uber-successful use of DBS to treat Parkinson disease has also shown.

---

## [Author Response]

The following is the authors' response to the original reviews.

**Reviewer #1 (Public Review):**
The authors investigated state-dependent changes in evoked brain activity, using electrical stimulation combined with multisite neural activity across wakefulness and anesthesia. The approach is novel, and the results are compelling. The study benefits from an in-depth sophisticated analysis of neural signals. The effects of behavioral state on brain responses to stimulation are generally convincing.It is possible that the authors' use of "an average reference montage that removed signals common to all EEG electrodes" could also remove useful components of the signal, which are common across EEG electrodes, especially during deep anesthesia. For example, it is possible (in fact from my experience I would be surprised if it is not the case) that under isoflurane anesthesia, electrical stimulation induces a generalized slow wave or a burst of activity across the brain. Subtracting the average signal will simply remove that from all channels. This does not only result in signals under anesthesia being affected more by the referencing procedure than during waking but also will have different effects on different channels, e.g. depending on how strong the response is in a specific channel.

We thank the reviewer for the positive comments and for raising this point. We do not believe that the average reference montage is obscuring an evoked slow wave in the isoflurane-anesthetized mice. Electrical stimulation did elicit a brief activation in nearby neurons that was followed by roughly 200 ms of quiescence, but no significant changes in firing in the other regions we recorded from (Author response image 1).

**Author response image 1. sa3fig1:** ERP and evoked population activity during isoflurane anesthesia do not show evidence of global responses. (Top). ERP (-0.2 to +0.8 s around stimulus onset) with all EEG electrode traces superimposed. Data represented is the same: red traces have been processed with the average reference montage, black traces have not. (Bottom) Population mean firing rates from the areas of interest from the same experiment as above.

We are familiar with the work from Dasilva et al. (2021), a study similar to ours because they also performed cortical electrical stimulation in mice anesthetized with isoflurane. They show widespread evoked multi-unit activity (derived from LFP) in isoflurane-anesthetized mice in response to electrical stimulation, but critical experimental differences may underlie the conflicting results presented in our study. Both works use similar levels of isoflurane to maintain anesthesia (we use a level roughly equivalent to their “deep” level). However, our experiments use only isoflurane, whereas Dasilva et al. induced anesthesia with ketamine and medetomidine followed by isoflurane. It has been shown that isoflurane and ketamine have different effects on neural dynamics (Sorrenti et al., 2021). Typically, isoflurane causes reduced spontaneous firing rates and decreased evoked response amplitudes compared to wakefulness, whereas ketamine has been shown to increase firing rates and evoked response amplitudes (Aasebø et al., 2017; Michelson & Kozai, 2018). Perhaps a more relevant difference are the electrical stimulation parameters used to perturb the brain. Dasilva et al. used 1 ms pulses of 500 μA, which would have a much larger effect than the stimulation used in this work, 0.2 ms pulses of 10-100 μA.

Additionally, we would like to clarify that the average reference montage is not impacting the main findings of this work. As the reviewer correctly pointed out, the average reference montage does change the appearance of the ERP in the butterfly plots (Top panel in Author response image 1). However, all the quantitative analyses of the EEG-ERPs are performed on the global field power, computed by taking the standard deviation across all EEG channels, which is not affected by the average reference montage.

**Reviewer #2 (Public Review):**
[…] The conclusions regarding the thalamic contributions to the ERP components are strongly supported by the data.The spatiotemporal complexity is almost a side point compared to what seems to be the most important point of the paper: showing the contribution of thalamic activity to some components of the cortical ERP. Scalp ERPs have long been regarded as purely cortical phenomena, just like most EEGs, and this study shows convincing evidence to the contrary.The data presented seemingly contradicts the results presented by Histed et al. (2009), who assert that cortical microstimulation only affects passing fibers near the tip of the electrodes, and results in distant, sparse, and somewhat random neural activation. In this study, it is clear that the maximum effect happens near the electrodes, decays with distance, and is not sparse at all, suggesting that not only passing fibers are activated but that also neuronal elements might be activated by antidromic propagation from the axonal hillock. This appears to offer proof that microstimulation might be much more effective than it was thought after the publication of Histed 2009, as the uber-successful use of DBS to treat Parkinson's disease has also shown.

We thank the reviewer for their positive comments and thoughtful suggestions. We appreciate and agree with the reviewer’s perspective that the thalamic contribution to the cortical ERP is one of the key points of this study. We also thank the reviewer for their comment on the apparently contradictory results reported by Histed et al. (2009). This gives us the opportunity to further highlight the important contribution of our study to the field.

First, we would like to highlight some key experimental differences between the two studies. In our study we used single pulse stimulation with currents between 10 and 100 μA, whereas Histed et al. used trains of pulses (100 ms in duration at 250 Hz) with lower current intensities (between 2 and 50 μA). We varied the depth of stimulation, targeting superficial and deep cortical layers; Histed et al. exclusively stimulated superficial cortical layers. In addition, the two studies used recording methods that are orthogonal in nature. We used Neuropixels probes that record from neurons that span all cortical layers depth-wise while Histed et al. used two-photon calcium imaging to record from a horizontal plane of neurons (again, in the superficial cortical layers).

Because of these important methodological differences, it is more appropriate to compare the Histed et al. results to our results from superficial stimulation at comparable current intensities. In this case, we believe the two studies show similar results: stimulation activated a small fraction of neurons even hundreds of microns away from the stimulating electrode (see Figure 4A from our manuscript). However, our study adds an important observation pointing to the critical role of the depth of the stimulating electrode. We observe significant excitation of local cortical neurons (Figure 4D) and trans-synaptic activation of the thalamus only when we delivered deep stimulation (Figure5A). This effect is likely mediated by activation of large, myelinated cortico-thalamic fibers, which are thought to be more excitable that non-myelinated horizontal fibers (Tehovnik & Slocum, 2013).

To summarize, Histed et al. (2009) concluded that microstimulation causes a sparse activation of a distributed set of neurons with little evidence of synaptically driven activation. Instead, we showed that microstimulation can robustly activate local neurons and trans-synaptically activate distant neurons when stronger stimuli are directed to deep cortical layers. Based on this, we conclude that electrical stimulation is indeed highly effective, and is a valid tool that can be used to probe and characterize the cortico-thalamo-cortical network of any behavioral state.

----------

**Reviewer**
*#*
**1 (Recommendations for the authors):**
1. I am not clear how "putative pyramidal" or RS and "putative inhibitory" fast-spiking neurons were identified. Please provide some further details on that, including average spike wave shapes, and distribution of firing rates, and it would be interesting to know the proportion of "putative" RS and FS neurons in your recorded population. Obviously, caution is warranted here because, without further work, you cannot be sure that those are indeed pyramidal cells or interneurons! Is this subdivision necessary at all?

We added details regarding the cell-type classification to the Results (lines 136-140) and the Methods section. This classification is common practice in cortical extracellular electrophysiology recordings given that cell-type specific analyses can reveal important differences between the two putative populations (Barthó et al., 2004; Bortone et al., 2014; Bruno & Simons, 2002; Jia et al., 2016; Niell & Stryker, 2008; Sirota et al., 2008). Based on our findings that the two populations respond to electrical stimulation in similar ways (excitation followed by a period of quiescence and rebound excitation), we agree the subdivision is not necessary to support our conclusions. However, we believe that some readers will appreciate seeing the two putative populations presented separately.

2. I wonder how the authors know whether the animals were awake, specifically when they were not running. Did you observe animals falling asleep when head-fixed? Providing some analyses of spontaneous EEG/LFP signals in each state could add some reassurance that only wakefulness was included, as intended.

While we cannot conclusively rule out that mice were asleep during the “quiet wakefulness” periods we analyzed, we believe they were likely to be awake for two main reasons: (1) all the experiments were performed during the dark phase of the light/dark cycle, when the mice are less likely to enter a sleep state (Franken et al., 1999); (2) the animals did not undergo specific training to promote drowsiness or sleep. Indeed, many sleep-focused studies in head-fixed mice are performed during the light phase of the animal’s cycle to maximize the likelihood of capturing sleep states (Kobayashi et al., 2023; Turner et al., 2020; Yüzgeç et al., 2018; Zhang et al., 2022). We have added this note to the Discussion section (lines 402-406).

Because we do not specifically record during sleep states and our recording does not include electromyography, which is commonly used in conjunction with EEG to classify sleep stages, we cannot accurately perform spectral comparison between “quiet wakefulness” and sleep states in our recordings.

3. I was unsure about the meaning of some of the terminology, specifically "rebound", "rebound spiking", "rebound excitation" etc. Why do you call it "rebound"?

“Rebound” is a term often used to describe a period of enhanced spiking following a period of prolonged silence or inhibition (Guido & Weyand, 1995; Roux et al., 2014). Grenier et al. list “postinhibitory rebound excitation” as an intrinsic property of cortical and thalamic neurons (1998). We added this description to the text (lines 79-80).

**Reviewer #2 (Recommendations For The Authors):**
Regarding analysis, I would make three main points:Regarding the CSD analysis, I think the authors have done a good job of circumventing several of the known issues of this technique, especially by using ERPs rather than ongoing activity. However, although I do not immediately have access to the literature to back up this claim, I've heard that many assumptions behind CSD require a laminar structure with electrodes positioned perpendicular to these layers. In Figure 1B it seems like the neuropixels probe is not really perpendicular to the cortical layers, and I wonder if this might be an issue. I am also wondering how to interpret the thalamic CSD, as this structure is not laminar, lacks the mass of neatly stacked neuronal dipoles present in the cortex, and does not have an orderly array of synaptic inputs and outputs. I understand that CSD analysis helps minimize the contributions of volume conduction, but in this case, I also wonder if the thalamic CSD is even necessary to back up the paper's claims.

One-dimensional CSD is computed assuming that the electrode is inserted perpendicular to cortex. This is mainly important for the interpretation of sinks and sources, since CSD can be also computed on radial voltages (e.g., EEG [Tenke & Kayser, 2012]). In general, our Neuropixels probes do not significantly deviate from perpendicular (mean deviation from perpendicular 15.3 degrees, minimum 5.2 degrees, and maximum 36.6 degrees). The probe represented in Figure 1B deviates from perpendicular by 31.2 degrees, which is an outlier compared to the rest of the insertions. Any deviation from perpendicular would result in the “effective” cortical thickness being larger by a factor of 1/cos(angle deviation from perpendicular) and thus would not affect the relative location of sources and sinks. We have added a statement to clarify this in the text (lines 126 and 454-456).

We agree with the statement regarding CSD analysis in the thalamus. We originally included the CSD for the thalamus in Figure 2F for completeness. As the reviewer pointed out, thalamic CSD was not used to perform any subsequent analysis and is, therefore, not necessary to back up any claims. As such, we have removed CSD plot from Figure 2F to avoid any confusion and made a comment to this effect in the legend (lines 1175-1177).

On the merits of using the z-score normalization for spike rates vs. other strategies like standardizing to maximum firing, I am aware that both procedures have limitations, but the z-score changes the range of the firing rate from [0, +Inf] to [-Inf, +Inf]. This does not seem correct considering that negative spiking rates do not exist. The standardization to maximum rate keeps the range within [0, 1], not creating negative rates. Another point that it will be worth discussing is the reported values of the z-scored values. For example, what does it mean to be 54 standard deviations away from the mean? 6 standard deviations is already a big distance from the mean.

For Figure 2, we chose to represent the neural firing rates as z-scores because we found it important to report the magnitude of both the increase and decrease of the evoked firing rates in the post-stimulus period relative to the pre-stimulus rate. The normalization we used helps to visualize the magnitude of the effects of electrical stimulation in neuronal activity for both directions, which is an important result of the study. Despite the differences between the two normalization methods, the normalization based on the maximum firing does not significantly change the qualitative interpretation of Figure 2 in the manuscript (Author response image 2).

**Author response image 2**

**Author response image 2. sa3fig2:** Evoked firing rates for neurons in the areas of interest in response to deep stimulation in MO during the awake state. (Left) Firing rates of all neurons normalized by the average, pre-stimulus firing rate. (Right) Firing rates of all neurons normalized by the maximum post-stimulus firing rate.

Regarding Figure 3 and the associated text, we would like to clarify that the magnitude metric is not simply a z-score value (with units of SD) but rather it is the integrated area under the z-scored response over the response window (with units of SD∙seconds). This can help explain why we see values of ~50 SD∙s. We chose to z-score firing rates, LFP, and CSD to normalize across the different signals and magnitudes of the evoked responses. We often observed the largest responses in the LFP (see Figure 3A), which may be partly due to the signal naturally having a larger dynamic range than the measured neural firing rates. Then we integrated the z-score response time series to capture the dynamic of the signal over the response window, rather than a static value such as the mean or maximum z-score. After performing a thorough literature search, we found no other ways to capture and compare the magnitudes of the different signals. We have added language to clarify the magnitude metric (lines 155-156) and added the appropriate units.

In reporting the p-values, I recommend increasing the number of significant digits to four because the p-value seems to be the same for different tests in several places (e.g.: lines 207 to 218), which seems odd. I also wonder whether this could be an artifact of the z-scoring procedure. In the figures, I would like to advise the use of 1 asterisk to denote "weak evidence to reject the null hypothesis (0.05 > p > 0.01)" and two asterisks to denote "strong evidence to reject the null hypothesis (0.01 > p)", and make a note of it accordingly in the manuscript and/or figure legends.

According to the reviewer’s suggestion, we have changed the statistics language to “* weak evidence to reject null hypothesis (0.05 > p > 0.01), ** strong evidence to reject null hypothesis (0.01 > p > 0.001), *** very strong evidence to reject null hypothesis (0.001 > p)” throughout the manuscript.

We have also increased the number of significant digits to four throughout the manuscript. It is true that some of the p-values reported for Figure 3 (lines 169-180) are the same for different tests. This is not an artifact of the z-scoring, but rather a consequence of performing the Wilcoxon signed-rank test (an ordinal statistical test) with small sample numbers. Because the p-value depends only on the relative ordering, not the continuous distribution of values, the small sample size (N=6-14) increases the likelihood of obtaining the exact same p-value if the relative ordering of samples is the same.

Line 202: If the magnitude corresponds to z-score data, please add "s.d." after the number, as z-scored values are expressed in standard deviation units. Please update this throughout the paper.

As stated above the magnitude metric is the integrated area under the z-scored response over the response window (with units of SD∙seconds). We have added the correct units in all places.

Line 214: Please report how the multiple comparisons correction was performed

We have added the test used for multiple comparisons in line 169 (formerly line 214) and in the Methods section (line 770).

Line 462: please replace "Neuropixels activity" with "LFP and single-unit activity".

We changed the wording to specify “LFP, and single neuron responses…” (now line 337).

Line 475: a short explanation of the bi-stability phenomena will be helpful for the reader.

We added the following description: “a state characterized by spontaneous alternation between bouts of activity and periods of silence” (lines 350-351).

Line 601: It is asserted that "Electrical stimulation directly activates local cells and axons that run near the stimulation site via activation of the axon initial segment" and the paper by Histed et al. 2009 is cited. This does not seem like an appropriate citation, as Histed et al. explicitly state that electrical microstimulation does not activate local neuronal bodies near the electrode tip. See my comment above.

Upon further reading, we believe we are seeing evidence of direct axonal activation and subsequent antidromic activation of local cell bodies, as you suggested in your above comment and has been proposed by many including Histed et al. (2009) and Nowak and Bullier (1998). We edited our sentence accordingly, kept the Histed et al. citation, and added other relevant citations (lines 487-490).

**References**

Aasebø, I. E. J., Lepperød, M. E., Stavrinou, M., Nøkkevangen, S., Einevoll, G., Hafting, T., & Fyhn, M. (2017). Temporal Processing in the Visual Cortex of the Awake and Anesthetized Rat. *ENeuro*, *4*(4), 59–76. https://doi.org/10.1523/ENEURO.0059-17.2017Barthó, P., Hirase, H., Monconduit, L., Zugaro, M., Harris, K. D., & Buzsáki, G. (2004). Characterization of Neocortical Principal Cells and Interneurons by Network Interactions and Extracellular Features. *Journal of Neurophysiology*, *92*(1), 600–608. https://doi.org/10.1152/jn.01170.2003Bortone, D. S., Olsen, S. R., & Scanziani, M. (2014). Translaminar Inhibitory Cells Recruited by Layer 6 Corticothalamic Neurons Suppress Visual Cortex. *Neuron*, *82*, 474–485. https://doi.org/10.1016/j.neuron.2014.02.021Bruno, R. M., & Simons, D. J. (2002). Feedforward Mechanisms of Excitatory and Inhibitory Cortical Receptive Fields. *The Journal of Neuroscience*, *22*(24), 10966–10975. https://doi.org/10.1523/JNEUROSCI.22-24-10966.2002Dasilva, M., Camassa, A., Navarro-Guzman, A., Pazienti, A., Perez-Mendez, L., Zamora-López, G., Mattia, M., & Sanchez-Vives, M. V. (2021). Modulation of cortical slow oscillations and complexity across anesthesia levels. *NeuroImage*, *224*, 117415. https://doi.org/10.1016/j.neuroimage.2020.117415Franken, P., Malafosse, A., & Tafti, M. (1999). Genetics of sleep regulation in mice-Franken et al Genetic Determinants of Sleep Regulation in Inbred Mice. *SLEEP*, *22*(2). https://academic.oup.com/sleep/article/22/2/155/2731698Grenier, F., Timofeev, I., & Steriade, M. (1998). Leading role of thalamic over cortical neurons during postinhibitory rebound excitation. *Proceedings of the National Academy of Sciences of the United States of America*, *95*(23), 13929–13934. https://doi.org/10.1073/pnas.95.23.13929Guido, W., & Weyand, T. (1995). Burst responses in thalamic relay cells of the awake behaving cat. *Journal of Neurophysiology*, *74*(4), 1782–1786. https://doi.org/10.1152/JN.1995.74.4.1782Histed, M. H., Bonin, V., & Reid, R. C. (2009). Direct Activation of Sparse, Distributed Populations of Cortical Neurons by Electrical Microstimulation. *Neuron*, *63*(4), 508–522. https://doi.org/10.1016/j.neuron.2009.07.016Jia, X., Siegle, J., Bennett, C., Gale, S., Denman, D. R., Koch, C., & Olsen, S. (2016). High-density extracellular probes reveal dendritic backpropagation and facilitate neuron classification 1 2. *Journal of Neurophysiology*, *121*(5), 1831–1847. https://doi.org/10.1101/376863Kobayashi, G., Tanaka, K. F., & Takata, N. (2023). Pupil Dynamics-derived Sleep Stage Classification of a Head-fixed Mouse Using a Recurrent Neural Network. *The Keio Journal of Medicine*, 2022-0020-OA. https://doi.org/10.2302/KJM.2022-0020-OAMichelson, N. J., & Kozai, T. D. Y. (2018). Isoflurane and ketamine differentially influence spontaneous and evoked laminar electrophysiology in mouse V1. *Journal of Neurophysiology*, *120*(5), 2232. https://doi.org/10.1152/JN.00299.2018Niell, C. M., & Stryker, M. P. (2008). Highly selective receptive fields in mouse visual cortex. *Journal of Neuroscience*, *28*(30), 7520–7536. https://doi.org/10.1523/JNEUROSCI.0623-08.2008Nowak, L. G., & Bullier, J. (1998). Axons, but not cell bodies, are activated by electrical stimulation in cortical gray matter. II. Evidence from selective inactivation of cell bodies and axon initial segments. *Experimental Brain Research*, *118*(4), 489–500. https://doi.org/10.1007/S002210050305/METRICSRoux, L., Stark, E., Sjulson, L., & Buzsáki, G. (2014). In vivo optogenetic identification and manipulation of GABAergic interneuron subtypes. *Current Opinion in Neurobiology*, *26*, 88–95. https://doi.org/10.1016/j.conb.2013.12.013Sirota, A., Montgomery, S., Fujisawa, S., Isomura, Y., Zugaro, M., & Buzsáki, G. (2008). Entrainment of Neocortical Neurons and Gamma Oscillations by the Hippocampal Theta Rhythm. *Neuron*, *60*(4), 683–697. https://doi.org/10.1016/j.neuron.2008.09.014Sorrenti, V., Cecchetto, C., Maschietto, M., Fortinguerra, S., Buriani, A., & Vassanelli, S. (2021). Understanding the Effects of Anesthesia on Cortical Electrophysiological Recordings: A Scoping Review. *International Journal of Molecular Sciences*, *22*(3), 1286. https://doi.org/10.3390/IJMS22031286Tehovnik, E. J., & Slocum, W. M. (2013). Two-photon imaging and the activation of cortical neurons. *Neuroscience*, *245*(March), 12–25. https://doi.org/10.1016/j.neuroscience.2013.04.022Tenke, C. E., & Kayser, J. (2012). Generator localization by current source density (CSD): Implications of volume conduction and field closure at intracranial and scalp resolutions. *Clinical Neurophysiology*, *123*(12), 2328–2345. https://doi.org/10.1016/J.CLINPH.2012.06.005Turner, K. L., Gheres, K. W., Proctor, E. A., & Drew, P. J. (2020). Neurovascular coupling and bilateral connectivity during nrem and rem sleep. *ELife*, *9*, 1. https://doi.org/10.7554/ELIFE.62071Yüzgeç, Ö., Prsa, M., Zimmermann, R., & Huber, D. (2018). Pupil Size Coupling to Cortical States Protects the Stability of Deep Sleep via Parasympathetic Modulation. *Current Biology*, *28*(3), 392. https://doi.org/10.1016/J.CUB.2017.12.049Zhang, X., Landsness, E. C., Chen, W., Miao, H., Tang, M., Brier, L. M., Culver, J. P., Lee, J. M., & Anastasio, M. A. (2022). Automated sleep state classification of wide-field calcium imaging data via multiplex visibility graphs and deep learning. *Journal of Neuroscience Methods*, *366*, 109421. https://doi.org/10.1016/J.JNEUMETH.2021.109421